# Self-Speculative Decoding Accelerates Lossless Inference in Any-Order and Any-Subset Autoregressive Models

**Gabe Guo**
Department of Computer Science
Stanford University
gabeguo@stanford.edu

**Stefano Ermon**
Department of Computer Science
Stanford University
ermon@cs.stanford.edu

## Abstract

In arbitrary-order language models, it is an open question how to sample tokens in parallel from the *correct joint distribution*. With discrete diffusion models, the more tokens they generate in parallel, the less their predicted distributions adhere to the originally learned data distribution, as they rely on a conditional independence assumption that only works with infinitesimally small timesteps. We find that a different class of models, any-subset autoregressive models (AS-ARMs), holds the solution. As implied by the name, AS-ARMs can generate tokens in any order, and in parallel. Moreover, AS-ARMs support parallelized joint probability density estimation, allowing them to correct their own parallel-generated token distributions, via our *Any-Subset Speculative Decoding (ASSD)* algorithm. ASSD provably enables generation of tokens from the correct joint distribution, with the number of neural network calls upper bounded by the number of tokens predicted – notably, previous speculative decoding algorithms lack our efficiency guarantee. We empirically verify that ASSD speeds up language generation, without sacrificing quality. Furthermore, we provide a mathematically justified scheme for training AS-ARMs for generation, and show that AS-ARMs achieve state-of-the-art performance among sub-200M parameter models on infilling benchmark tasks, and nearly match the performance of models 50X larger on code generation. Our theoretical and empirical results indicate that the once-forgotten AS-ARMs are a promising direction of language modeling.

## 1 Introduction

Almost all the SoTA LLMs (Achiam et al., 2023; Touvron et al., 2023; Liu et al., 2024a; Team et al., 2023) are autoregressive, *i.e.*, they only support left-to-right token generation. As a result, they suffer from two major problems: (1) they must generate tokens one-by-one, which limits their speed; (2) they cannot infill sequences in orders besides left-to-right (unless specialized training strategies are adopted (Bavarian et al., 2022; Fried et al., 2022; Roziere et al., 2023), but these are heuristic and not guaranteed to output the correct structure).

Regarding models that inherently support infilling, there are discrete diffusion models (Austin et al., 2021; Campbell et al., 2022; Lou et al., 2023; Sahoo et al., 2024) and any-order autoregressive models (AO-ARMs) (Yang, 2019; Shih et al., 2022). Discrete diffusion models have the benefit of parallel sampling of multiple tokens at a time, which potentially speeds up generation, but at the cost of fidelity to the learned data distribution (Lou et al., 2023). On the other hand, fast sampling schemes have generally not been explored for AO-ARMs.

We explore the problem of fast parallel sampling from AO-ARMs, without any degradation in output quality. Particularly, we are inspired by speculative decoding techniques, which have accelerated generation in standard autoregressive models by using a draft model to quickly generate multiple tokens. Then, tokens are accepted or rejected based on their fidelity to the joint distribution as evaluated by the oracle model. Interestingly, the outputted distribution is provably the same as would have been obtained from sampling only from the expensive oracle model, while usually (but not

always) using fewer neural network forward passes (Chen et al., 2023; Leviathan et al., 2023). Crucial to speculative decoding are (1) density estimation from the oracle, and (2) a fast draft model. Indeed, AO-ARMs, by design, estimate joint probability density of sequences (Yang, 2019). Furthermore, AO-ARMs can even act as their own draft models; due to their architectural design and training objective, they can generate tokens in any order and in parallel (Yang, 2019; Shih et al., 2022).

As such, we propose *Any-Subset Speculative Decoding (ASSD)*, an algorithm that combines speculative decoding with AO-ARMs. ASSD provably generates sequences from the true joint distribution learned by the oracle model. Our method's chief advantages over previous speculative decoding algorithms include: (1) It is *mathematically guaranteed to never increase the number of function evaluations (i.e., language model calls); in practice, this means that it speeds up generation without losing quality (as we empirically verify).* In contrast, vanilla speculative decoding has no such guarantee on NFEs, and could slow down runtime in some cases. (2) ASSD can handle $O(2^N)$ possible infilling tasks, exponentially more than the $O(N)$ allowed by traditional speculative decoding. (3) We get the speculator "for free", without the need to train an auxiliary draft model.

We also provide a mathematically justified training scheme for AO-ARMs. Finally, we show that appropriately trained AO-ARMs achieve state-of-the-art performance among sub-200M parameter models (diffusion and autoregressive) on infilling benchmark tasks, and nearly match the performance of models 50X larger on code generation, while needing fewer finetuning tokens.

## 2 BACKGROUND

**Autoregressive Models:** Given a text sequence $\mathbf{x} \sim \mathcal{D}$, where $\mathcal{D}$ is a data distribution, autoregressive (AR) models learn

$$p(\mathbf{x}) = \prod_{i=0}^{|\mathbf{x}|-1} p(x_i|x_0, \ldots, x_{i-1}).$$
(1)

The product rule factorization means that an AR model only needs to learn conditional distributions with support of size $O(S)$, where $S$ is the size of the vocabulary. This factorization also admits a straightforward generation strategy, where token $x_i$ is sampled from $p(\cdot|x_0, \ldots, x_{i-1})$, *i.e.*, the previous tokens are used to produce the next (Achiam et al., 2023). This also means that the prompt for an AR model must always be the prefix, *i.e.*, $\mathbf{x}_{0:i}$ – arbitrarily-located prompts are not supported.

**Infilling:** We consider infilling tasks, where the prompt is not necessarily the prefix, but can be arbitrarily-located. Examples of this are code generation, story completion, and scientific data imputation. Mathematically, this can be formulated as sampling from a joint conditional probability distribution with discrete state space. That is, we wish to sample from $p(\mathbf{x}_{\sigma(\geq m)}|\mathbf{x}_{\sigma(<m)})$, where $\sigma(i)$ is the (zero-indexed) positional index of the $i$-th ordered item in the sequence of length $N$. In other words, $\sigma$ is a permutation of $\{0, 1, \ldots, N-1\}$, and $i$ is the generation order. So, $\mathbf{x}_{\sigma(<m)}$ represents the prompt tokens, and $\mathbf{x}_{\sigma(\geq m)}$ represents the tokens whose distributions we want to predict. In general, we can have any $\sigma : \{0, \ldots, N-1\} \to \{0, \ldots, N-1\}$, so long as $\sigma$ is a bijection. As previously established, regular AR models cannot address this task in general, except when $\sigma(i) = i$.

**Any-Order Autoregressive Models:** Any-order autoregressive models (AO-ARMs) (Shih et al., 2022; Hoogeboom et al., 2021; Yang, 2019) can be seen as collections of $N!$ joint distributions indexed by the factorization order $\sigma$:

$$\log p(\mathbf{x}_{\sigma(\geq m)}|\mathbf{x}_{\sigma(<m)}) = \sum_{i=m}^{N-1} \log p(x_{\sigma(i)}|\mathbf{x}_{\sigma(<i)}; \sigma).$$
(2)

That is, the distribution for each token is calculated one at a time, in order of increasing $i$, and used as conditioning for the next token to be predicted. Concretely, each evaluation of the network produces a probability distribution (for a single token) with support of size $O(S)$, where $S$ is the size of the vocabulary. The probability of each member of the support is explicitly calculated and stored in memory, incurring $O(S)$ memory cost per token.

If trained to optimality, all the joint distributions should be equal. However, except with infinite data and capacity, given different ordering functions $\alpha$ and $\sigma$ (Shih et al., 2022),

$$\sum_{i=0}^{N-1} \log p(x_{\sigma(i)}|\mathbf{x}_{\sigma(<i)};\sigma) \neq \sum_{i=0}^{N-1} \log p(x_{\alpha(i)}|\mathbf{x}_{\alpha(<i)};\alpha). \tag{3}$$

**Any-Subset Autoregressive Models:** Any-subset autoregressive models (AS-ARMs) are a subclass of AO-ARMs that reduce the number of joint distributions ($\sigma$) learned from $N!$ to $2^N$. This puts less training burden on the finite model capacity, while keeping strictly the same expressivity as it relates to conditional joint distributions of the form in Equation 2. To achieve this, AS-ARMs adopt the recursive binary lattice mask decomposition protocol from (Shih et al., 2022). The idea underlying this protocol is that we can split every generation task into two parts: the prompt (denoted by $\mathbf{x}_{\sigma(<m)}$) and the tokens that need to be generated (denoted by $\mathbf{x}_{\sigma(\geq m)}$). Mathematically, we want to estimate $p(\mathbf{x}_{\sigma(\geq m)}|\mathbf{x}_{\sigma(<m)})$, where $m$ is the number of tokens in the prompt (assuming 0-indexing). Now, we make two observations.

Firstly, since we never need to evaluate the density of the conditioning $\mathbf{x}_{\sigma(<m)}$, we have every token attend to every other token within $\mathbf{x}_{\sigma(<m)}$. Secondly, within $\mathbf{x}_{\sigma(\geq m)}$, there is no need to learn all the possible factorization paths, as long as we can get the joint conditional probability $p(\mathbf{x}_{\sigma(\geq m)}|\mathbf{x}_{\sigma(<m)})$. Taking inspiration from vanilla autoregressive models, we enforce

$$\forall\, i \geq m, j \geq m : \sigma(i) > \sigma(j) \Leftrightarrow i > j. \tag{4}$$

That is, we simply process the masked tokens left to right. Thus, given the location of the prompt, we only have to learn one path to calculate the joint probability of the generation. This also solves the inconsistency problem in Equation 3. (As later shown, this is crucial to Algorithm 1's correctness.)

This reduces the number of queries learned by the model from $N!$ (all possible permutations) to $2^N$ (all possible mask location selections, times one ordering per mask selection). This makes optimization easier, as noted by (Shih et al., 2022) and verified in our ablations (Figure 3). In summary, AS-ARMs are a subclass of AO-ARMs incorporating Equation 4's disambiguated ordering strategy. The architecture design is typically the same, but the way we query it is different.

## 3   Sampling Strategies for Joint Distributions

We want to accurately and efficiently sample from the joint conditional distribution in Equation 2. We assume that we have access to AS-ARMs that explicitly predict single-variable marginals, *i.e.*, next-token prediction under some ordering.

**One-Step Sampling from the Joint:** Sampling directly from this joint distribution in a single step is typically infeasible, because its support has size $O(S^{N-m})$, where $S$ is the number of tokens in the vocabulary. If we wanted to explicitly calculate the probability of every tuple in the support, the exponential space cost to just store the distribution would quickly exceed memory capacities.

**Sequential Sampling via Factorization:** We can sample the $\log p(x_{\sigma(i)}|\mathbf{x}_{\sigma(<i)})$ one-by-one, using each generated token as the conditioning for the next one. Following Equation 2, we can get samples from the joint conditional distribution by doing this $N - m$ times, once for each $i \in [m, N)$. In (any-order, any-subset) autoregressive models (Achiam et al., 2023; Touvron et al., 2023; Yang, 2019; Shih et al., 2022), there is typically $O(S)$ time cost per-token, so the overall time cost would be $O(S * (N - m))$. Indeed, this is the dominant generation strategy for autoregressive models like GPT (Achiam et al., 2023), where $\sigma(i) = i$.

**Parallel Sampling via Independence Assumption:** At another extreme, we can independently sample multiple single-variable marginals in parallel. That is, we predict $\log p(x_{\sigma(i)}|\mathbf{x}_{\sigma(<m)})$ for all $i \in [m, N)$. Since $N - m$ generations are done in parallel, computational time cost is only $O(S)$, which is effectively constant with respect to the number of tokens. The limitation is that

$$\sum_{i \in [m, N)} \log p(x_{\sigma(i)}|\mathbf{x}_{\sigma(<m)}) \neq \log p(\mathbf{x}_{\sigma(\geq m)}|\mathbf{x}_{\sigma(<m)}), \tag{5}$$

except in the unlikely scenario of true independence, *i.e.*, $\log p(x_{\sigma(i)}|\mathbf{x}_{\sigma(<m)}) = \log p(x_{\sigma(i)}|\mathbf{x}_{\sigma(<i)})$.

This is analogous to how discrete diffusion models take large discretized timesteps in the reverse CTMC to predict tokens in parallel, even though the predictions at each position are actually generated with a conditional independence assumption that only holds for an infinitesimally small timestep (*i.e.*, one-by-one generation) (Lou et al., 2023; Sahoo et al., 2024).

**Best of Both Worlds: Any-Subset Speculative Decoding:** We seek a way to combine the runtime benefits of parallel sampling with the fidelity of sequential sampling. That is, can we achieve $O(S)$ time complexity for arbitrary-subset generation, while recovering true samples from $\log p(\mathbf{x}_{\sigma(\geq m)}|\mathbf{x}_{\sigma(<m)})$? The key insight here is that if we had an oracle model that could evaluate the joint density of a sequence with a singular function evaluation, we could leverage the quick speed of independent parallel generation and use these samples $\log p(x_{\sigma(i)}|\mathbf{x}_{\sigma(<m)})$ as estimates for the true $\log p(x_{\sigma(i)}|\mathbf{x}_{\sigma(<i)})$. Then, via some rejection sampling scheme which uses as an oracle the joint density evaluation of this newly generated sequence, we could keep only the samples that adhere to $\log p(\mathbf{x}_{\sigma(\geq m)}|\mathbf{x}_{\sigma(<m)})$. In principle, this could take best-case $o(S)$ time (parallelized), if the oracle accepts all the samples, and worst-case $O(S*(N-m))$ time.

Speculative decoding is one such method that guarantees fidelity to the true distribution. It is mathematically proven to generate samples from the target distribution, and the empirical results show a stark decrease in the number of function evaluations required (Leviathan et al., 2023; Chen et al., 2023). However, prior to our work, speculative decoding has only been shown to work for left-to-right (Achiam et al., 2023) autoregressive models.

# 4 ARCHITECTURAL DESIGN OF AS-ARMS

Towards the goal of fast, principled parallel sampling in any order, we seek any-subset autoregressive models (AS-ARMs) that can support our *Any-Subset Speculative Decoding (ASSD)* algorithm (which is fully described in Section 5). To recap, the criteria are: (1) generates arbitrarily-ordered tokens in parallel; (2) evaluates joint density with only one forward pass. We now describe how AS-ARMs/AO-ARM architectures should be designed to fulfill these criteria.

**Parallel Sampling via Arbitrary Positional Queries:** With one function evaluation, we should be able to simultaneously predict in parallel (conditionally independent) distributions for all the masked tokens $\mathbf{x}_{\sigma(\geq m)}$, conditioned on the prompt $\mathbf{x}_{\sigma(<m)}$. This allows the network to act as a quick "draft" model. Concretely, we should be able to pass in arbitrary positional queries $\sigma(\geq m)$ of not-yet-predicted tokens to condition on the visible prompt tokens $\mathbf{x}_{\sigma(<m)}$. As implied by the "any-order" moniker, there is no constraint on which positions we query: we could query the leftmost, rightmost, or even a randomly selected unfilled position. Furthermore, the positional queries all attend to the same prompt tokens, but the prompt tokens cannot attend to the positional queries. So, no matter how many positions we query in parallel, each query cannot change the representations of the prompt tokens, and therefore cannot change the outcomes of the other simultaneous queries. See Figure 1a.

**Density Estimation via Causal-Like Attention Masking:** Another crucial ingredient of speculative decoding is density estimation – this allows the "oracle" model to correct the mistakes of the draft model. As such, discrete diffusion models trained with an ELBO (Lou et al., 2023; Sahoo et al., 2024; Deschenaux & Gulcehre, 2024) are not readily adaptable to this scheme. In contrast, the training objective of AO-ARMs/AS-ARMs (Shih et al., 2022; Yang, 2019) teaches them to evaluate joint densities of sequences.

Care must be taken, however, to pick an architecture that evaluates the joint density of a sequence in one function evaluation, *i.e.*, $O(S)$ time. Some architectures (Shih et al., 2022; Hoogeboom et al., 2021) take $O(S*N)$ steps, as only logits of masked tokens are predicted at each function evaluation – they are unable to predict the logits of visible tokens. To get an architecture that can calculate the logits for all visible and masked tokens in a single function evaluation with $O(S)$ time, we can design the attention masks for each token to only allow attention (*i.e.*, conditioning) to the preceding tokens in the ordering. Then, we can construct a factorization as in Equation 2. We process tokens in parallel with this attention mask, giving us the desired $O(S)$ time. Mathematically,

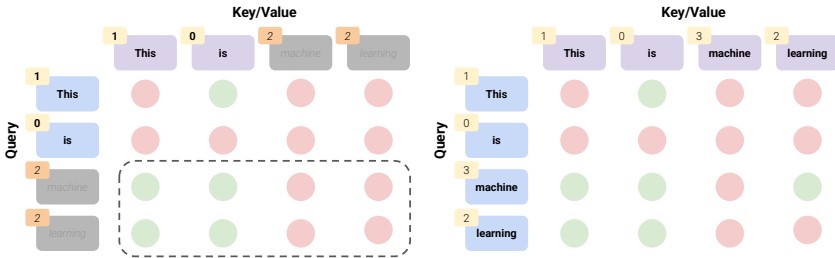

(a) **Parallel Sampling Attention Mask**    (b) **Density Estimation Attention Mask**

Figure 1: **Attention Masks in AS-ARMs:** Red means that attention is not allowed, while green means that attention is allowed from the query to the key/value. The numbers on each token represent the generation order, where lower-numbered tokens come first. Panel 1a shows how, by attending to the same conditioning ("This", "is"), the tokens "machine" and "learning" can be generated in parallel. They do not attend to themselves nor each other. Panel 1b shows how we can conduct one-step density estimation on a sequence, with a permuted causal-like attention mask. The mask enforces the order "is", "This", "learning", "machine", where each token only attends to those decoded before it.

$$A_{\sigma(i),\sigma(j)} = \begin{cases} 0 & i \le j \\ 1 & i > j \end{cases}, \tag{6}$$

where $A_{\sigma(i),\sigma(j)}$ is the masking matrix for the attention maps – $0$ means that the query token at index $\sigma(i)$ is *not* allowed to attend to the key/value token at index $\sigma(j)$; $1$ means that token $\sigma(i)$ *can* attend to token $\sigma(j)$. Such an attention mask can yield faithful estimates of $\log p(\mathbf{x}_{\sigma(\ge i)}|\mathbf{x}_{\sigma(<i)})$. See Figure 1b, and Appendix D for further discussion.

**Two-for-One Model: Streamlined Drafting:** Ideally, we do not want to train a separate draft model, because it takes extra memory on the hardware and additional training cost. Furthermore, the computations from the separate draft model cannot necessarily be re-used for the oracle. If the draft model was the same as the target/oracle model, we would not incur extra memory nor training cost, and could cache computations from the draft model to accelerate the target calculations.

**Suitable Architectures:** Many popular architectures actually satisfy the aforementioned constraints, as they essentially boil down to being able to specify flexible attention masks according to the desired factorization of the distribution. One suitable architecture family is XLNet encoders (Yang, 2019) (discussed further in Appendix D). GPT-style decoders (Achiam et al., 2023; Touvron et al., 2023) can also be minimally modified with specialized positional encodings to become suitable AS-ARM architectures (Pannatier et al., 2024). We run the following experiments with XLNet.

## 5 How to Modify Speculative Decoding for AS-ARMs?

We present *Any-Subset Speculative Decoding (ASSD)*, and prove some of its properties. See Algorithm 1. See Appendix A for proofs. In the resampling step, we use the notation $(f(x))_+ = \frac{\max(0,f(x))}{\sum_x \max(0,f(x))}$.

**Lemma 1.** *The first token speculated in each loop iteration will always be accepted. That is, Line 19's conditional always evaluates to true when $i = n$.*

**Theorem 1.** *Algorithm 1 requires no more than $N - m$ total function evaluations of $p(\cdot|\cdot)$. That is, there will never be more calls to a neural network than the number of tokens returned on Line 28.*

**Theorem 2.** *Algorithm 1 produces samples from the true joint distribution $p(\mathbf{x}_{\sigma(\ge m)}|\mathbf{x}_{\sigma(<m)})$.*

Based on Lemma 1 and Theorem 1 (see proof), we should always set $k > 2$ (where $k$ is the number of speculated tokens per call to the draft model). We also present a variant of ASSD in Appendix E.4 with a context-derived n-gram as the draft model (Stewart et al., 2024). However, this variant does not fulfill Lemma 1.

**Comparison to Speculative Decoding:** Superficially, ASSD's instructions are similar to vanilla speculative decoding (Chen et al., 2023; Leviathan et al., 2023). However, our ASSD actually has

---

**Algorithm 1:** Any-Subset Speculative Decoding

---

**Input:** $k$: number parallel tokens; $N$: target sequence length; $\sigma$: mapping of decoding order to 0-based positional index; $p(\cdot|\cdot)$: any-order autoregressive model; $\mathbf{x}_{\sigma(<m)}$: prompt tokens

**Output:** $\mathbf{x}_{\sigma(\geq m)}$: the predicted tokens

---

1   $n \leftarrow m$ // number of tokens we've already decoded

2   **while** $n < N$ **do**

3     $t \leftarrow \min(n+k, N)$

4     // speculate the next $t$ tokens

5     **(Parallelized): for** $i \in [n:t)$ **do**

6       $\tilde{x}_{\sigma(i)} \sim p(\cdot|\mathbf{x}_{\sigma(<n)})$ // sample from partially conditioned distribution

7       $p_{\sigma(i)} \leftarrow p(\tilde{x}_{\sigma(i)}|\mathbf{x}_{\sigma(<n)})$ // get partially conditioned density

8     **end**

9     **if** $n == T - 1$ **then**

10       $x_{\sigma(n)} \leftarrow \tilde{x}_{\sigma(n)}$ // accept the proposal

11       **return** $\mathbf{x}_{\sigma(\geq m)}$

12     **end**

13     **(Parallelized): for** $i \in [n:t)$ **do**

14       $q_{\sigma(i)} \leftarrow p(\tilde{x}_{\sigma(i)}|\mathbf{x}_{\sigma(<n)}, \tilde{\mathbf{x}}_{\sigma[n:i)})$ // get ground truth density

15     **end**

16     // rejection sampling

17     **for** $i \in [n:t)$ **do**

18       $r \sim \mathcal{U}[0,1]$

19       **if** $r < \min(1, \frac{q_{\sigma(i)}}{p_{\sigma(i)}})$ **then**

20         $x_{\sigma(i)} \leftarrow \tilde{x}_{\sigma(i)}$ // accept the proposal

21       **else**

22         $x_{\sigma(i)} \sim \left(p(\cdot|\mathbf{x}_{\sigma(<n)}, \tilde{\mathbf{x}}_{\sigma[n:i)}) - p(\cdot|\mathbf{x}_{\sigma(<n)})\right)_{+}$ // resample

23         **exit** from for loop

24       **end**

25     **end**

26     $n \leftarrow i + 1$ // update number of decoded tokens

27   **end**

28   **return** $\mathbf{x}_{\sigma(\geq m)}$

---

an *important theoretical guarantee that vanilla speculative decoding lacks*, in that Theorem 1 upper bounds the NFEs (draft + oracle) to the number of tokens generated. In contrast, vanilla speculative decoding has no such guarantee: Leviathan et al. (2023) note that if the draft model is particularly poor and/or expensive, vanilla speculative decoding can theoretically increase the total NFEs and therefore runtime (particularly when the drafts keep getting rejected). Furthermore, our algorithm has exponentially larger capacity in infilling patterns it can handle: $O(2^N)$, while vanilla speculative decoding can only handle $O(N)$ patterns, due to the any-subset versus left-to-right natures of the algorithms. Finally, we do not require auxiliary draft models, as they come for free with AS-ARMs.

## 6   TRAINING AND IMPLEMENTATION

**Architectural Instantiation:** We use the 110M parameter case-sensitive version of XLNet from Huggingface (Wolf et al., 2019), which is one of many architectures that satisfies the criteria in Section 4. We finetune starting from the pretrained weights (Yang, 2019).

**Dataset**: We finetune on the OpenWebText dataset (Gokaslan et al., 2019). Following (Sahoo et al., 2024), we pack the sequences together, and split them into chunks of 512 tokens, based on XLNet's case-sensitive tokenizer with a vocabulary of $32,000$ possible tokens. We have separator tokens to delineate the start of a new document.

**Training Hyperparameters:** The original XLNet (Yang, 2019) was only trained to predict 85 masked tokens in a sequence length of 512, corresponding to less than $20\%$ of the sequence, which is

not ideal for generative modeling tasks. We want a model that can predict tokens almost from scratch, so we finetune the off-the-shelf XLNet model. More hyperparameter details are in Appendix E.

**Teacher-Forced Joint Loss:** For our training objective, we maximize the joint conditional probability in Equation 2 with cross-entropy loss:

$$\max_\theta \mathbb{E}_{m \sim f(\cdot), \sigma \sim s(\cdot|m)} \left[ \log p_\theta(\mathbf{x}_{\sigma(\geq m)} | \mathbf{x}_{\sigma(<m)}) \right], \tag{7}$$

where $f(\cdot)$ is a distribution over integers from $[0, N)$ (*i.e.*, sampling prompt length $m$), and $s(\cdot|m)$ is a distribution of permutations of integers from $[0, N)$ conditioned on prompt length $m$ (*i.e.*, sampling token ordering $\sigma$), where $N$ is the sequence length. The loss has three major components: (1) joint conditional distribution; (2) expectation over token orderings; (3) expectation over prompt lengths.

***Joint Conditional Objective:*** To justify the joint conditional distribution $\log p(\mathbf{x}_{\sigma(\geq m)} | \mathbf{x}_{\sigma(<m)})$, assume $m$ and $\sigma$ are fixed. We define a discrete time (absorbing state) Markov chain $\mathbf{x}, \mathbf{x}_{\sigma(<N-1)}, \mathbf{x}_{\sigma(<N-2)}, \ldots, \mathbf{x}_{\sigma(<m+1)}, \mathbf{x}_{\sigma(<m)}$, with time index $t \in \{0, 1, 2, \ldots, N-m-1, N-m\}$, as in Figure 2. That is, $X_t = \mathbf{x}_{\sigma(<N-t)}$. To generate data, we follow the time reversal of this Markov chain. To obtain the time reversal, first consider reversing a singular time step, from $t$ to $t-1$. This corresponds to learning

$$\log p_\theta(X_{t-1} | X_t) = \log p_\theta(\mathbf{x}_{\sigma(<N-(t-1))} | \mathbf{x}_{\sigma(<N-t)}) = \log p_\theta(\mathbf{x}_{\sigma(N-t+1)} | \mathbf{x}_{\sigma(<N-t)}). \tag{8}$$

That is, we predict the next token's density. To reverse the whole process, we sum for each timestep:

$$\sum_{t=1}^{N-m} \log p_\theta(\mathbf{x}_{\sigma(<N-t+1)} | \mathbf{x}_{\sigma(<N-t)}) = \log p_\theta(\mathbf{x}_{\sigma(<N)} | \mathbf{x}_{\sigma(<m)})$$
$$= \log p_\theta(\mathbf{x}_{\sigma(<m)}, \mathbf{x}_{\sigma(\geq m)} | \mathbf{x}_{\sigma(<m)}) = \log p_\theta(\mathbf{x}_{\sigma(\geq m)} | \mathbf{x}_{\sigma(<m)}), \tag{9}$$

which gives us Equation 7's joint conditional probability. This loss is different than the conditionally independent losses used in (Shih et al., 2022) and discrete diffusion models (Sahoo et al., 2024; Lou et al., 2023). Notably, their architectures, due to the lack of causal-like attention masking, could not support joint losses.

***Expectations over Token Orderings and Prompt Lengths:*** In the objective (Equation 7), we do not make assumptions about the distribution of the prompt length $m \sim f(\cdot)$ nor the distribution of the prompt ordering $\sigma \sim s(\cdot|m)$ (so long as $\sigma$ follows the decomposition protocol laid out in Equation 4. In general, these distributions are task-dependent. For instance, in regular autoregressive tasks, $\sigma$ would be deterministic, *i.e.*, the identity function.

For this paper, we are interested in generating text from near-scratch, so we train a model where $m \sim \mathcal{U}[0.01 * N, 0.10 * N]$. To get $\sigma$, we first sample $\sigma_{\text{pre}} \sim \mathcal{U}(S_N)$, where $S_N$ represents all the possible permutations of the integers from $[0, N)$. But, remembering Equation 4's efficient masking protocol, we sort each of $\sigma_{\text{pre}}(< m)$ and $\sigma_{\text{pre}}(\geq m)$ in ascending order to get $\sigma$. This eliminates the ambiguity in paths the model has to learn to calculate a given joint, while maintaining which positions are in the prompt and which need to be predicted. Appendices E.1 and G contain more details.

## 7 EXPERIMENTS

Our first experiment (Section 7.1) empirically verifies that ASSD with AS-ARMs indeed preserves the output distribution while being faster, as predicted by our theoretical analysis. Our other experiments (Section 7.2) show that on both natural language and coding benchmarks, AS-ARMs, even with a fraction of the training resources, beat other model classes of comparable size, and are competitive with models orders of magnitude larger.

### 7.1 CORRECTNESS AND SPEED OF ANY-SUBSET SPECULATIVE DECODING

We prompt the model with 640 masked sequences from the WikiText test dataset (Merity et al., 2016). As in training, sequences are packed together into chunks of 512 tokens. We randomly mask out 95% of each sequence, leaving 5% of tokens (uniformly scattered throughout the sequence) as the prompt. We evaluate our finetuned model. See Table 1 for results. Appendix L shows sample outputs.

| Sampler | Gen PPL | Entropy | Model NFE | Aux NFE | Time (s) |
|---|---|---|---|---|---|
| *Sequential* | $107.9 \pm 1.6$ | $7.65 \pm 0.01$ | $486.0 \pm 0.0$ | $\mathbf{0.0 \pm 0.0}$ | $18.21 \pm 0.00$ |
| *ASSD (N-Gram)* | $111.7 \pm 2.0$ | $7.64 \pm 0.01$ | $\mathbf{422.0 \pm 0.7}$ | $422.0 \pm 0.7$ | $16.80 \pm 0.03$ |
| ***ASSD (Self)*** | $107.6 \pm 1.6$ | $7.64 \pm 0.01$ | $\underline{434.1} \pm 0.4$ | $\mathbf{0.0 \pm 0.0}$ | $\mathbf{16.50 \pm 0.02}$ |

Table 1: **Comparison of Speculative and Sequential Decoding:** Left-to-right, entries show mean and standard error of generative perplexity (judge: GPT-2 Large), Shannon entropy, number of AS-ARM function evaluations, number of auxiliary draft model calls, and wall clock time. *ASSD (Self)* is from Algorithm 1. We also modify Algorithm 1 with context-derived *N-Grams* (Stewart et al., 2024) as a draft model (see Appendix E.4). We set $k = 5$ for ASSD.

| Model | Size | Code Tokens | Lang. Tokens | Pass @ 1 |
|---|---|---|---|---|
| XLNet-Code | **110M** | **15B** | **0B** | 38.59 |
| DiffuLLaMA | 6738M | 19B | 46B | **40.68** |

Table 2: **Performance on HumanEval Infilling:** Comparison of code infilling abilities of XLNet finetuned on code versus DiffuLLaMA (Gong et al., 2024). We use HumanEval's single-line infilling task (Bavarian et al., 2022), evaluated by pass@1 (each attempt counts, instead of only the best attempt) on 5165 trials.

**Empirical Correctness of Distribution:** As expected from Theorem 2, the output distribution from speculative decoding is statistically the same as the output distribution from sequential decoding, measured by generative perplexity and entropy.

**Speed-Up:** Finally, both variants of ASSD (parallel sampling from self as draft, context n-gram (Stewart et al., 2024) as draft) provide a statistically significant speedup over regular sequential decoding, as predicted by Lemma 1. (When decoding sequentially, the number of calls to the neural network is the same as the number of masked tokens.) ASSD with paralllel sampling is the fastest method, and requires the least total NFEs. ASSD with context n-gram is not that far behind: the intuition is that although it gives lower-quality drafts, it is very cheap. This was also observed in similar studies for AR models (Stewart et al., 2024). However, only $1.15$ tokens get generated per iteration when using context n-gram, as opposed to $2.24$ tokens with parallel sampling as draft.

## 7.2 INFILLING BENCHMARK TASKS

**Code Generation:** In coding, bidirectional context is crucial, making an appealing use case for AS-ARMs. Table 2 shows that AS-ARMs finetuned on code are competitive with models that are orders of magnitude larger. Furthermore, AS-ARMs use fewer finetuning tokens. We evaluate against DiffuLLaMA as a representative baseline, as other discrete diffusion and autoregressive models of comparable size were shown to be non-competitive (Gong et al., 2024). See Appendix E.6.

| Model | Size | Tokens | Infill 1/5 | | Infill 3/5 | |
|---|---|---|---|---|---|---|
| | | | *ROUGE 1/2/L* | *NFE* | *ROUGE 1/2/L* | *NFE* |
| GPT2-S | 127M | n/a[1] | 9.5/0.4/8.7 | $10.7 \pm 2.8$ | 13.5/0.6/10.2 | $31.8 \pm 6.0$ |
| SEDD-S | 170M | 210B | 11.6/0.8/10.7 | $32.0 \pm 0.0$ | 16.2/1.3/12.2 | $64.0 \pm 0.0$ |
| MDLM | 130M | 262B | 11.6/1.1/10.7 | $32.0 \pm 0.0$ | 13.3/1.0/10.4 | $64.0 \pm 0.0$ |
| DiffuGPT-S | 127M | +130B | 14.0/1.5/13.0 | $32.0 \pm 0.0$ | 16.4/**2.0**/**14.2** | $64.0 \pm 0.0$ |
| *AS-ARM-PT* | ***110M*** | *2T[1]* | **14.4**/**1.7**/**13.1** | $\mathbf{8.7 \pm 2.5}$ | 7.7/0.6/6.4 | $\mathbf{18.5 \pm 6.8}$ |
| ***AS-ARM-FT*** | ***110M*** | ***+12B*** | 13.1/1.1/12.0 | $10.4 \pm 2.8$ | **18.0**/1.4/13.2 | $\underline{30.4} \pm 6.0$ |

Table 3: **Performance on ROCStories Infilling:** We test on 1871 short stories of five sentences each, and get 5 completions (trials) per story. We compare ROUGE ($\uparrow$). "Tokens" is the number of training tokens (excluding those for the pretrained initialization). "Infill 1/5" inputs sentences $\{1, 2, 4, 5\}$ and infills $\{3\}$. "Infill 3/5" inputs sentences $\{1, 5\}$ and infills $\{2, 3, 4\}$. We report $\mu \pm \sigma$ NFEs. "+" indicates additional finetuning tokens, after the pretrained initialization. See footnote 1.

**Natural Language:** We also investigate whether AS-ARMs can outperform discrete diffusion models and regular autoregressive models on natural language infilling tasks, in which bidirectional context is key. We follow Gong et al. (2024)'s setup on ROCStories (Mostafazadeh et al., 2016). Table 3 shows that AS-ARMs are generally better than the baselines, while using the fewest parameters and finetuning resources [1]. AS-ARM-PT (pretrained XLNet weights (Yang, 2019) from Huggingface, without finetuning) is the best at infilling a single missing sentence. This corresponds to a $\sim 20\%$ masking ratio, which is roughly what it was pretrained on (see Appendix E.2). Finetuning on a wider distribution of masking ratios as in Equation 7 splits finite model capacity among multiple tasks. Unsurprisingly, when infilling three out of five sentences, our AS-ARM-FT (finetuned AS-ARM) surpasses all the other models on ROUGE-1, and is behind only DiffuGPT-S on ROUGE-2 and ROUGE-L. *Overall, AS-ARMs are best on four out of six metrics in Table 3.*

## 7.3 OTHER EXPERIMENTS

See Appendices G and F for more experiments, including: ablation on number of speculated tokens, extension to sequential benchmark tasks, ablation on sequence length, ablation on acceptance threshold, KV caching, increased model size, training setup ablation.

## 8 RELATED WORKS

**Any-Order Autoregressive Models:** The defining characteristic of any-order autoregressive models is their ability to estimate the probability density of arbitrary marginal and/or conditional queries. This generalizes vanilla autoregressive models. We primarily base our work off of XLNet (Yang, 2019) and MAC (Shih et al., 2022). From XLNet, we adopt the idea of causal attention: crucially, this allows us to estimate probability density in a parallelized single step, *i.e.*, $O(1)$ time (Yang, 2019). From MAC, we adopt recursive decomposition of queries on a binary lattice, which reduces the permutations needed to be learned from $N!$ to $2^N$ (Shih et al., 2022): this made MAC the first any-subset autoregressive model.

More recent AO-ARMs (Hoogeboom et al., 2021; Strauss & Oliva, 2021; Shih et al., 2022) abandoned causal attention, in favor of full attention with absorbing state tokens to represent "masked" queries. This increases runtime of joint density estimation from $O(1)$ to $O(N)$: to satisfy the factorization rule, each token's density is estimated one-by-one so it conditions only on the preceding tokens.

**Discrete Diffusion:** Discrete diffusion models data generation as the reversal of an inhomogeneous continuous-time Markov chain (CTMC). Empirically, the best performing CTMC is an absorbing state process. Discrete diffusion can generate tokens in parallel, with arbitary prompt locations. The seminal work is D3PM (Austin et al., 2021). Later, LDR (Campbell et al., 2022) formalized discrete diffusion under the lens of CTMC theory. More recently, SEDD (Lou et al., 2023), MDLM (Sahoo et al., 2024), DiffuLLaMA (Gong et al., 2024), and LLaDA (Nie et al., 2025) created discrete diffusion models that rivaled or surpassed autoregressive models.

One limitation of discrete diffusion models is their inability to do joint density estimation. Also, although tokens can be generated in parallel with large discretized timesteps, they are generated under the conditional independence assumption, which may not adhere to the limiting joint distribution. Furthermore, the architectures perform full attention across all tokens, even the masks (Lou et al., 2023; Sahoo et al., 2024); due to this lack of causal attention masking, it is not straightforward to apply KV-caching.

**Speculative Decoding:** Speculative decoding is an algorithm developed for autoregressive models that enables quick token generation from an inexpensive draft model. Via some techniques related to rejection sampling, the outputs provably adhere to the joint distribution of the target language model (Leviathan et al., 2023; Chen et al., 2023). As compared to other speculative decoding works, our method has the advantages of: (1) theoretical guarantee to *never* increase NFEs; (2) provides the speculator "for free" without training any auxiliary models; (3) addresses the any-subset infilling

---

[1]To our knowledge, the pretraining details for the public weights of XLNet-Base (which we call AS-ARM-PT: pretrained) were never released. *If* we assume that XLNet-Base uses the same pretraining hyperparameters as those reported for XLNet-Large, it would have been trained on around two trillion tokens (Yang, 2019). GPT-2 also did not release their full training recipe (Radford et al., 2019).

problem, which has $O(2^N)$ tasks, exponentially more than the $O(N)$ left-to-right tasks addressed in other works. *Detailed comparison to other related works are in Appendix H.*

## 9 CONCLUSION

Our results suggest that the long-forgotten AS-ARMs are a promising language modeling methodology. We theoretically and empirically show that the longstanding problem of principled parallel token generation from the joint distribution is easily solved once we combine AS-ARMs' parallelized density estimation and generation with speculative decoding. We can essentially get the speedup as a "free lunch", theoretically guaranteed to come without degradation in output quality (as opposed to parallel decoding methods in discrete diffusion models). Furthermore, benchmark tasks indicate that AS-ARMs can match or surpass the currently dominant language modeling paradigms. Future work includes scaling to billion-parameter models.

## REPRODUCIBILITY STATEMENT

Beyond the main text, Appendix E contains details on experimental setup. Models available at https://huggingface.co/therealgabeguo/ASARM. Code available at https://github.com/gabeguo/any-order-speculative-decoding.

## ETHICS STATEMENT

Our work does not have any ethical concerns beyond the usual concerns associated with generative language modeling research.

## ACKNOWLEDGEMENTS

This material is based upon work supported by the U.S. Department of Energy, Office of Science, Office of Advanced Scientific Computing Research, Department of Energy Computational Science Graduate Fellowship under Award Number DE-SC0025528 to Gabe Guo. We thank Amil Merchant, Tristan Saidi, Luke Bailey, and Ajay Sridhar for helpful discussion.

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

## A  PROOFS

Line numbers refer to Algorithm 1.

**Lemma 1.** *The first token speculated in each loop iteration will always be accepted.*

*Proof.* When $i = n$ on Line 13,

$$q_{\sigma(i)} = p(\tilde{x}_{\sigma(i)}|\mathbf{x}_{\sigma(<n)}, \tilde{\mathbf{x}}_{\sigma[n:i]}) \tag{10}$$

$$= p(\tilde{x}_{\sigma(i)}|\mathbf{x}_{\sigma(<n)}, \tilde{\mathbf{x}}_{\sigma[n:n]}) \tag{11}$$

$$= p(\tilde{x}_{\sigma(i)}|\mathbf{x}_{\sigma(<n)}) \tag{12}$$

$$= p_{\sigma(i)}. \tag{13}$$

So, $r < \frac{q_{\sigma(i)}}{p_{\sigma(i)}} = 1$ on Line 19, and thus, when $i = n$, $\tilde{x}_{\sigma(n)}$ will always be accepted on Line 20. □

**Theorem 1.** *Algorithm 1 requires no more than $N - m$ total function evaluations of $p(\cdot|\cdot)$. That is, there will never be more calls to a neural network than the number of tokens decoded.*

*Proof.* The idea underlying this proof is that each iteration of the while loop on Line 2 has two function evaluations. The first function evaluation is on Lines 5 to 8, to speculate tokens. The second function evaluation is on Lines 13 to 15, to evaluate the oracle density.

We then just need to show that at least two tokens will be decoded, making for minimum one token generated per function evaluation.

By Lemma 1, the first token (at $i = n$) is always accepted. So, we can move on to $i = n + 1$.

When $i = n + 1$, regardless of whether we accept or reject $\tilde{x}_{\sigma(n+1)}$, we will still obtain a value for $x_{\sigma(n)}$, whether it is $\tilde{x}_{\sigma(n)}$ on Line 20 or a resampling from the adjusted $()_+$ distribution on Line 22 (which does not require additional calls to $p(\cdot|\cdot)$, as it makes use of already calculated distributions from Lines 6 and 14).

So, on each loop iteration, we are guaranteed to get token values for at least $x_{\sigma(n)}, x_{\sigma(n+1)}$, thereby giving us the requisite two tokens.

On the last loop iteration, if $n = N - 1$, there is an edge case where only one token can be decoded (Line 9). But, as previously shown in Lemma 1, this speculated token is mathematically guaranteed to be accepted, since it was the first and only one to be speculated. So, we can forgo the verification step for this final loop iteration. Thus, to generate this final token, we still only need one function evaluation, maintaining the lower bound of one token generated per function evaluation. □

**Theorem 2.** *Algorithm 1 produces samples from the true joint distribution $p(\mathbf{x}_{\sigma(\geq m)}|\mathbf{x}_{\sigma(<m)})$.*

*Proof.* The only differences between our algorithm and speculative decoding are:

1. The use of $p$ as its own speculator. This is valid, because speculative decoding is agnostic to the speculator function, as long as it gives probability density estimates.

2. The omission of the verification step when decoding the last token $n = T - 1$ on Line 9. This is justified by Lemma 1, which shows that when $i = n = T - 1$, $\tilde{x}_{\sigma(n)}$ would always meet the acceptance criteria if it were to go through the verification on Line 19.

3. We do not sample an extra token if all of the speculations are accepted at the end of each loop iteration. However, this does not affect the distribution of the other tokens outputted. We skip this extra token sampling because, as shown in Lemma 1, the first token speculated in the next iteration will always be accepted, which "makes up" for the omission of the extra sample.

Other than these differences, this algorithm is simply speculative decoding, but with a different ordering than $\sigma = [0, 1, 2, \ldots, N-1]$. We can permute/sort $\mathbf{x}$ in increasing order of $\sigma(i)$, and define as the result $\mathbf{y}$. Then, our algorithm is mathematically equivalent to speculative decoding with the inputs $\mathbf{y}$ as the sequence, and the draft model the same as the target model. The rest of the proof of correctness of our algorithm is therefore the same as in (Leviathan et al., 2023; Chen et al., 2023). For completeness, we reproduce their proof in Appendix B, with some additional commentary. $\quad\square$

## B  MODIFIED REJECTION SAMPLING CORRECTNESS

We provide an extended proof of correctness for Theorem 2, as it relates to the modified rejection sampling step in Algorithm 1. Note that the main ideas and layout of this proof are copied from (Chen et al., 2023) and (Leviathan et al., 2023). We only include it here for the reader's convenience and consistency with our notation.

*Proof.* WLOG, consider the random variable (token) $x_{\sigma(i)}$, returned by Algorithm 1 (*ASSD*). Consider the iteration where $n \leq i < n + k \leq N$, *i.e.*, the iteration that generates $x_{\sigma(i)}$.

For the algorithm to be correct, we desire that $\mathbb{P}(x_{\sigma(i)} = x) = p(x_{\sigma(i)} = x|\mathbf{x}_{\sigma(<i)})$, where $\mathbb{P}(x_{\sigma(i)} = x)$ is the probability that *ASSD* generates token value $x$ at position $\sigma(i)$, and $p(x_{\sigma(i)} = x|\mathbf{x}_{\sigma(<i)})$ is the probability that regular sequential decoding would generate $x$ at position $\sigma(i)$. In other words, the distribution of *ASSD* is the same as in sequential decoding. Note that $\mathbf{x}_{\sigma(<i)} = \mathbf{x}_{\sigma(<n)} \oplus \tilde{\mathbf{x}}_{\sigma[n:i]}$ in both sequential decoding and our algorithm, because they both use generations from previous iterations (in addition to the prompt) as conditioning for future iterations.

Let $\tilde{x}$ (shorthand for $\tilde{x}_{\sigma(i)}$) be the token value generated from the conditionally independent draft in Line 6. When $x_{\sigma(i)} = x$ is true, there are two mutually exclusive and collectively exhaustive possibilities (from the if-else statement): (1) $\tilde{x}$ was accepted, and $\tilde{x} = x$ (2) $\tilde{x}$ was rejected, and $x$ was the result of the resampling.

$$
\begin{aligned}
\mathbb{P}(x_{\sigma(i)} = x) &= \mathbb{P}(\tilde{x} \text{ accepted}, \tilde{x} = x) + \mathbb{P}(x_{\sigma(i)} = x, \tilde{x} \text{ rejected}) \\
&= \mathbb{P}(\tilde{x} \text{ accepted } |\tilde{x} = x)\mathbb{P}(\tilde{x} = x) + \mathbb{P}(\tilde{x} \text{ rejected})\mathbb{P}(x_{\sigma(i)} = x|\tilde{x} \text{ rejected})
\end{aligned}
\tag{14}
$$

Analyzing the first term, we look to the "if" (accept) clause on Line 19:

$$
\begin{aligned}
\mathbb{P}(\tilde{x} \text{ accepted } |\tilde{x} = x)P(\tilde{x} = x) &= \min\left(1, \frac{q_{\sigma(i)}(x)}{p_{\sigma(i)}(x)}\right) p_{\sigma(i)}(x) \\
&= \min\left(p_{\sigma(i)}(x), q_{\sigma(i)}(x)\right)
\end{aligned}
\tag{15}
$$

Here, $q_{\sigma(i)}(x) = p(x_{\sigma(i)} = x|\mathbf{x}_{\sigma(<n)}, \tilde{\mathbf{x}}_{\sigma[n:i]})$, *i.e.*, the oracle density from Line 14; and $p_{\sigma(i)}(x) = p(x_{\sigma(i)} = x|\mathbf{x}_{\sigma(<n)})$, *i.e.*, the speculator density from Line 7.

Regarding the second term, we look at the "else" (reject) clause on Line 21. We analyze each item in the product separately, as the expressions are long:

$$
\begin{aligned}
\mathbb{P}(\tilde{x} \text{ rejected}) &= 1 - \mathbb{P}(\tilde{x} \text{ accepted}) \\
&= 1 - \sum_{x'} \mathbb{P}(x_{\sigma(i)} = x', \tilde{x} \text{ accepted}) \\
&= 1 - \sum_{x'} \min\left(p_{\sigma(i)}(x'), q_{\sigma(i)}(x')\right) \\
&= \sum_{x'} q_{\sigma(i)}(x') - \sum_{x'} \min\left(p_{\sigma(i)}(x'), q_{\sigma(i)}(x')\right) \\
&= \sum_{x'} q_{\sigma(i)}(x') - \min\left(p_{\sigma(i)}(x'), q_{\sigma(i)}(x')\right) \\
&= \sum_{x'} \max\left(q_{\sigma(i)}(x') - p_{\sigma(i)}(x'), q_{\sigma(i)}(x') - q_{\sigma(i)}(x')\right) \\
&= \sum_{x'} \max\left(q_{\sigma(i)}(x') - p_{\sigma(i)}(x'), 0\right)
\end{aligned}
\tag{16}
$$

From Line 22:

$$\mathbb{P}(x_{\sigma(i)} = x | \tilde{x} \text{ rejected}) = \big(p(x_{\sigma(i)} = x | \mathbf{x}_{\sigma(<n)}, \tilde{\mathbf{x}}_{\sigma[n:i)}) - p(x_{\sigma(i)} = x | \mathbf{x}_{\sigma(<n)})\big)_+$$
$$= \big(q_{\sigma(i)}(x) - p_{\sigma(i)}(x)\big)_+ \tag{17}$$
$$= \frac{\max\big(q_{\sigma(i)}(x) - p_{\sigma(i)}(x), 0\big)}{\sum_{x'} \max\big(q_{\sigma(i)}(x') - p_{\sigma(i)}(x'), 0\big)}$$

Following Equations 16 and 17, we have

$$\mathbb{P}(\tilde{x} \text{ rejected})\mathbb{P}(x_{\sigma(i)} = x | \tilde{x} \text{ rejected}) = \max\big(q_{\sigma(i)}(x) - p_{\sigma(i)}(x), 0\big) \tag{18}$$

Putting it all together back into Equation 14,

$$\mathbb{P}(x_{\sigma(i)} = x) = \min\big(p_{\sigma(i)}(x), q_{\sigma(i)}(x)\big) + \max\big(q_{\sigma(i)}(x) - p_{\sigma(i)}(x), 0\big)$$
$$= q_{\sigma(i)}(x)$$
$$= p(x_{\sigma(i)} = x | \mathbf{x}_{\sigma(<n)}, \tilde{\mathbf{x}}_{\sigma[n:i)}) \tag{19}$$
$$= p(x_{\sigma(i)} = x | \mathbf{x}_{\sigma(<i)})$$

The last line of Equation 19 is true, because to have gotten to index $i$ in the accept-reject loop (Line 17), we must have accepted $\tilde{\mathbf{x}}_{\sigma[n:i)}$, *i.e.*, $\mathbf{x}_{\sigma[n:i)} = \tilde{\mathbf{x}}_{\sigma[n:i)}$. Therefore, we have shown that *ASSD* gives the same per-token distribution as in sequential decoding. Induction over $i \in [m : N)$ will easily show that *ASSD* gives the correct joint distribution.

$\square$

## C  PROBABILISTIC GRAPHICAL MODEL

See Figure 2 for a probabilistic graphical model of the data generation process.

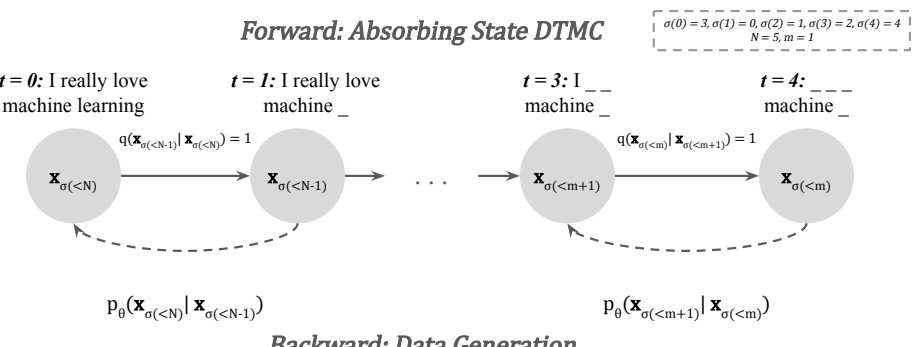

Figure 2: **Probabilistic Graphical Model:** Shows the discrete-time Markov chain for the forward noising process, and its time reversal (*i.e.*, data generation). This justifies Equation 7.

## D  CAUSAL-LIKE ATTENTION MASKING

We provide further discussion on the masking introduced in Section 4.

In particular, there is a subtlety in attention mechanisms: even if a token does not attend to itself, it still "sees" its own content because it constructs a query representation to compare against the keys. At first glance, this seemingly breaks the ban (Equation 6) on a token attending to itself. The solution is to separate the positional and content information into two streams, as done in XLNet (Yang, 2019).

The positional stream (which only has positional representations as input) is used to calculate the queries, *i.e.*, row index of the attention map. The content stream (which has token value information) is used to calculate the key/value information, *i.e.*, column index of the attention map. Thus, the evaluation of the positional queries does not "cheat" by looking at the ground truth content at its own position. Conceptually, this can be said to be more like cross-attention than the self-attention common in discrete diffusion models (Austin et al., 2021).

In contrast, architectures commonly used for discrete diffusion models (Lou et al., 2023; Deschenaux & Gulcehre, 2024; Sahoo et al., 2024; Austin et al., 2021) and other any-order autoregressive models (Shih et al., 2022; Hoogeboom et al., 2021) do not support this kind of masking. In these architectures,

$$\forall i, j : A_{\sigma(i),\sigma(j)} = 1, \tag{20}$$

which means that every token is allowed to attend to every token including itself when calculating its probability. From a probabilistic graphical modeling perspective, the outputted probabilities very roughly correspond to $\log p(x_{\sigma(i)}|\mathbf{x})$, which would *not* give us principled estimates on the already-visible tokens.

## E ADDITIONAL EXPERIMENTAL DETAILS

### E.1 MASK DISTRIBUTION

We also use a similar low-discrepancy sampler as (Sahoo et al., 2024) to reduce variance among prompt lengths within a batch.

### E.2 TRAINING HYPERPARAMETERS

In finetuning XLNet, we use maximum learning rate $10^{-4}$ and batch size 320 (16 per device, 4 accumulations, 5 devices). We have linear learning rate warmup for 5000 steps, and linear learning rate decay for $70,000$ additional steps, making for a total of $2.4 * 10^7$ samples seen. (Notably, this is far fewer than the $2.56 * 10^8$ samples that DiffuGPT-S saw (Gong et al., 2024).) We start at $15\%$ masking rate, then linearly increase the minimum masking rate to $90\%$ and the maximum masking rate to $99\%$ over 5000 steps.

We train on NVIDIA RTX A6000 Ada devices, which have 48 GB of RAM each. Training lasts around four days. We calculate validation loop metrics every 500 steps on 64 samples. We use the AdamW (Loshchilov et al., 2017; Kingma & Ba, 2014) optimizer.

We also tried training for more epochs, but we found that performance saturated after $\sim 10$B training tokens.

### E.3 METRICS

We are focused on generation quality. To that end, we have two metrics.

To quantify how likely the generated outputs of our model are, we calculate generative perplexity, where lower values are better. It is calculated as

$$\text{PPL}_{\text{gen}} = \left(e^{\sum_i \log q(x_i|\mathbf{x}_{0\ldots i-1})}\right)^{-1/N}, \tag{21}$$

where $q$ is an oracle language model (we use GPT-2 Large (Radford et al., 2019)), and $\mathbf{x}$ is a generated sequence.

To quantify the diversity in tokens, we calculate Shannon entropy (Shannon, 1948), where higher values are better. The formula is

$$H(\mathbf{x}) = -\sum_{x_i \in \mathbf{x}} \log_2(p(x_i))p(x_i), \tag{22}$$

where $p(x_i) = \frac{|\{j|\mathbf{x}_j = x_i\}|}{|\mathbf{x}|}$, *i.e.*, the frequency of a token in the sequence.

Typically, there is a trade-off between generative perplexity and Shannon entropy – highly repetitive sequences of common words like "a" have low generative perplexity, but also low entropy. Random sequences of gibberish have high entropy and high generative perplexity. We want to generate sequences with high entropy and low generative perplexity.

### E.4 SPECULATIVE DECODING VARIANTS

We compare sequential decoding to two variants of ASSD: the first is described in Algorithm 1. The second uses context-based n-grams, which were initially proposed for left-to-right speculative decoding (Stewart et al., 2024), although they easily fit into our framework. To adopt it to arbitrary order, we replace the speculator $p(\cdot|\cdot)$ in Lines 5-8 of Algorithm 1 with a context-based n-gram model $c(\cdot|\cdot)$, as in Algorithm 2. There, $c(a|b)$ is the probability over the partially decoded sequence that a bigram starting in $b$ ends in $a$, as in Equation 23:

$$c(a|b) = \frac{\sum_{i,x_i \neq \text{MASK}, x_{i+1} \neq \text{MASK}} \mathbb{1}[\mathbf{x}_{i,i+1} = (a,b)]}{\sum_{j,x_j \neq \text{MASK}} \mathbb{1}[x_j = b]}, \tag{23}$$

We initialize $c(\cdot|\cdot)$ by sweeping over the prompt, then update it iteratively, as the sequence is decoded.

---

**Algorithm 2:** Speculation with Context-Based N-Grams

**Input:** same as Algorithm 1
**Output:** see Algorithm 1
1 **for** $i \in [n:t)$ **do**
2      **if** $x_{\sigma(i)-1} \neq \textit{MASK}$ **then**
3          $x_{\text{cond}} \leftarrow x_{\sigma(i)-1}$
4      **else**
5          $x_{\text{cond}} \leftarrow \tilde{x}_{\sigma(i)-1}$
6      **end**
7      $\tilde{x}_{\sigma(i)} \sim c(\cdot|x_{\text{cond}})$ // sample from partially conditioned distribution
8      $p_{\sigma(i)} \leftarrow c(\tilde{x}_{\sigma(i)}|x_{\text{cond}})$ // get partially conditioned density
9 **end**

---

**Theorem 3.** *When $i \geq 1$, Algorithm 2 always sets $x_{cond}$ to a valid non-MASK value.*

*Proof.* We conduct strong induction on $i \geq m$, where $m$ is the given prompt length.

For a given value of $i$, we have two cases:

1. $x_{\sigma(i)-1} \neq \text{MASK}$. This case is trivial.

2. $x_{\sigma(i)-1} = \text{MASK}$. This reduces to showing that $\tilde{x}_{\sigma(i)-1} \neq \text{MASK}$. Firstly, there exists some $j$ such that $\sigma(j) = \sigma(i) - 1$, which trivially means $\sigma(j) < \sigma(i)$. From Equation 4, this is equivalent to saying $j < i$. By the inductive hypothesis, $\tilde{x}_{\sigma(j)} \neq \text{MASK}$ should have been speculated for all $j < i$.

$\square$

We note that context-based n-grams lack the guarantees of Lemma 1, so they could increase the total number of function evaluations (AS-ARMs + n-gram draft) beyond the sequence length. However, n-grams are typically cheap to evaluate, which could make up for the increased function evaluations.

### E.5 INFILLING BENCHMARK

We run $5$ trials over the dataset of $1871$ stories, making for $9355$ samples per masking level (short or long) per model. The models compared are: GPT-2 (Radford et al., 2019), MDLM (Sahoo et al., 2024), SEDD (Lou et al., 2023), DiffuGPT (Gong et al., 2024), AS-ARM-PT (pretrained XLNet weights) (Yang, 2019), and AS-ARM-FT (finetuned by us, starting from Huggingface initialization). We do not compare against models like LLaDA (Nie et al., 2025) because they have access to significantly more training budget than us, not to mention their orders-of-magnitude larger model size.

For the AS-ARM models, we use ASSD. We set $k = 15$ for our speculative decoding. Without speculative decoding on AS-ARMs, "Infill 1/5" requires $10.9\pm2.9$, and "Infill 3/5" requires $32.4\pm6.2$

| Sampler | Gen PPL | Entropy | NFEs | Time (s) |
|---|---|---|---|---|
| *Sequential* | $59.08 \pm 5.61$ | $3.119 \pm 0.065$ | $486.0 \pm 0.0$ | $18.04 \pm 0.00$ |
| *Speculative* | $59.12 \pm 5.26$ | $3.206 \pm 0.064$ | $247.3 \pm 1.8$ | $9.36 \pm 0.07$ |
| *Difference* | $+0.08\%$ | $+2.78\%$ | $-49.12\%$ | $-48.09\%$ |

Table 4: **Comparison of ASSD (Algorithm 1) and Sequential Decoding in Off-the-Shelf Model:** The entries show mean and standard error of generative perplexity (judge: GPT-2 Large), Shannon entropy, number of network function evaluations, and wall clock time. Metrics are calculated over 640 decoded WikiText sequences of length 512, where 95% is randomly masked out. We set $k = 5$ in speculative decoding.

evaluations. For GPT, we use sequential decoding, since it cannot be its own speculator. Following (Gong et al., 2024), we only give GPT the left conditioning, as it is not straightforward to give it rightward conditioning without instruction-tuning. This experiment can be run on a 16GB GPU.

## E.6 CODE GENERATION

We finetune our AS-ARM on 14.7 billion tokens from Starcoder's Python data (Li et al., 2023). This corresponds to $75,000$ training steps with a batch size of $384$ (16 per device, 4 accumulation steps, 6 GPUs) of 512 tokens each. We started from learning rate 0, warmed up for $20,000$ steps to learning rate $1.2 * 10^{-4}$, then scheduled to decay linearly for $63,333$ steps (which corresponds to $32 * 10^6$ total samples seen), although the run crashed slightly earlier than this. We start from a 5% masking rate, then warm up over $20,000$ steps to a 15% minimum masking rate, and a 99% maximum masking rate, sampled uniformly.

Since the XLNet tokenizer does not support whitespaces, we replace whitespace tokens with special characters during training (\n : <cls>, \t : <sep>, "_ _" : <unk>, "_ _ _" : <pad>), where _ is the space. When generating code, we reverse the special character mapping and truncate the leading space from the generated lines, since the tokenizer by default inserts a leading space to each word.

We evelute on the HumanEval single-line infilling task on 1033 Python test cases, following (Gong et al., 2024). We generated 5 completions for each test case, for a sample size of 5165. We evaluate with the pass@1 metric, *i.e.*, we count the failure or success of each of the five completions for a test case (rather than taking the best out of the five completions).

For the baseline, we compare the publicly available DiffuLLaMA (Gong et al., 2024) model. We find that the results from (Gong et al., 2024) are actually under-reported. When manually inspecting the outputs, we found that there were often generations that were correct, except that the number of leading spaces was off by one. Our investigation suggests that the LLaMA 7B tokenizer also seems to have an issue with counting prefix spaces, etc. Since this is not a problem with model capacity, we relaxed the evaluation, and manually inserted the ground truth indentation (*i.e.*, correct number of leading spaces) to DiffuLLaMA's outputs. After doing so, the pass@1 rate increased from about 16% to 40%.

This experiment can be run on a 16GB GPU.

## F ADDITIONAL RESULTS

### F.1 ANY-SUBSET SPECULATIVE DECODING: COMPARISON OF FINETUNED AND OFF-THE-SHELF

See Table 4. This shows extended results from Table 1, with the off-the-shelf model from Huggingface.

While the generative perplexity of the off-the-shelf model is much lower than the generative perplexity of our finetuned model, this comes at the cost of very low entropy. Indeed, when we manually inspected the outputs, we found that the off-the-shelf model generated highly repetitive, nonsensical sequences of a few common words (see Appendix K). On the other hand, the finetuned model

generated sentences that were, for the most part, coherent both semantically and syntatically (see Appendix L).

Additionally, the off-the-shelf model gains a larger boost in runtime from speculative decoding. Examining the outputted sequences, it again appears to be due to its highly repetitive output distributions, as these would be easier to speculate with a mean field model.

## F.2 How Many Tokens to Speculate?

| k | 2 | 3 | 4 | 5 | 6 | 8 | 10 | 15 | 20 |
|---|---|---|---|---|---|---|----|----|----|
| **Length: 128** | | | | | | | | | |
| **Speedup (%)** | -4.0 | 6.7 | 8.8 | 8.9 | 9.1 | 8.7 | 8.4 | 9.0 | 9.0 |
| **PPL p-Value** | 0.66 | 0.37 | 0.22 | 0.49 | 0.12 | 0.41 | 0.48 | 0.22 | 0.83 |
| **Entropy p-Value** | 0.14 | 0.20 | 0.73 | 0.48 | 1.00 | 0.28 | 0.76 | 0.52 | 0.17 |
| **Length: 256** | | | | | | | | | |
| **Speedup (%)** | -2.7 | 6.6 | 8.4 | 8.8 | 9.3 | 8.9 | 8.4 | 8.5 | 8.2 |
| **PPL p-Value** | 0.31 | 0.03 | 0.32 | 0.13 | 0.47 | 0.74 | 0.77 | 0.39 | 0.78 |
| **Entropy p-Value** | 0.43 | 0.60 | 0.49 | 0.15 | 0.17 | 0.48 | 0.56 | 0.95 | 0.22 |

Table 5: **Tuning k:** Generally, ASSD is not sensitive to the value of $k$ (number of tokens speculated per call to draft model), so long as it is at least $4$.

We have ablations on the effect of $k$ in Table 5. Generally, our selected value of $k = 5$ was a close-to-optimal choice. The difference in perplexity and entropy, according to t-tests, is also not statistically significant (see p-values), as predicted by Theorem 2 (which does not depend on $k$). Results in the below tables are calculated from 750 sequences, where we infill 95% of the tokens per sequence of length 128 and 256.

As we had predicted in Section 5, a choice of $k = 2$ slowed down the algorithm, since the algorithm, no matter what, is guaranteed to generate at least two tokens (with two function evaluations: speculate and verify) per loop iteration. So, if only two tokens are speculated, there could never be a speedup over the sequential capabilities, and in fact, the overhead time from running speculative decoding would slow it down.

## F.3 Effect of Sequence Length

| Length | Speedup (%) | PPL p-value | Entropy p-value |
|--------|-------------|-------------|-----------------|
| 128 | 8.94 | 0.49 | 0.48 |
| 256 | 8.84 | 0.13 | 0.15 |
| 512 | 9.44 | 0.71 | 0.96 |
| 640 | 10.45 | 0.23 | 0.24 |
| 768 | 10.59 | 0.80 | 0.35 |
| 896 | 10.90 | 0.94 | 0.48 |
| 1024 | 11.00 | 0.78 | 0.50 |

Table 6: **Effect of Sequence Length on Speedup:** Set $k = 5$, infill 95% of the sequence.

As we increase the sequence length, the speedup becomes more pronounced. The differences in output distribution from sequential decoding are not statistically different, as predicted by Theorem 2. See Table 6.

## F.4 KV-Cache

We implement KV caching. The prompt (which has full self-attention) is fixed. As we generate new tokens, they do *not* become part of the prompt; that way, we can keep their causal attention and not have to recompute the prompt hidden states, while still caching the new token representations. We do recompute positional encodings, since XLNet uses relative positional encodings (Yang, 2019).

We generally found that KV caching was not helpful for smaller model sizes and sequence lengths. But, it does help as we scale up the problem size. When using a 340M parameter XLNet (off-the-shelf, as we did not have the resources to retrain) to infill 128 tokens out of sequence length 1024 (as opposed to 110M parameter on sequence length 512), we observe that KV caching yields a 50% speedup in model inference time. See Table 7.

| Caching | Time (s) |
|---------|----------|
| KV Cache | $0.0706 \pm 0.0006$ |
| No Cache | $0.1407 \pm 0.0004$ |

Table 7: **KV Cache Benefits:** Average inference time of the model (per parallel speculation NFE), averaged over 100 trials (KV cache, versus not using KV cache). Results on 340M-parameter XLNet to infill 128 out of 1024 tokens.

### F.5 EPSILON-TOLERANCE

On line 19 (Algorithm 1), we tried adding a small absolute tolerance to the $\frac{q_{\sigma(i)}}{p_{\sigma(i)}}$ acceptance threshold, such that the decision is $r < \min(1, \frac{q_{\sigma(i)}}{p_{\sigma(i)}} + \epsilon)$. Intuitively, having higher tolerance will lead to more speculations being accepted, at the cost of quality.

See Table 8. We do 500 trials filling in 95% of a 256 token sequence, with $k = 5$. (They were run with different model weights than in the other experiments, but given that the results align with our expectations, we don't think they would change much on different model weights.) Basically, while we could get a little more speed boost by increasing the error threshold, the generative perplexity gets a lot worse.

| $\epsilon$ Tolerance | Speedup (%) | Spec PPL | Spec Entropy | Seq PPL | Seq Entropy | PPL p-Val | Entropy p-Val |
|---|---|---|---|---|---|---|---|
| 0 | 8.28 | $122.70 \pm 2.85$ | $6.98 \pm 0.01$ | $127.31 \pm 3.13$ | $6.99 \pm 0.01$ | 0.20 | 0.32 |
| $1 * 10^{-4}$ | 8.46 | $120.72 \pm 2.61$ | $6.97 \pm 0.01$ | $121.76 \pm 2.76$ | $6.98 \pm 0.01$ | 0.77 | 0.39 |
| $1 * 10^{-3}$ | 8.30 | $123.28 \pm 3.40$ | $6.97 \pm 0.01$ | $120.84 \pm 2.97$ | $6.97 \pm 0.01$ | 0.54 | 0.67 |
| $1 * 10^{-2}$ | 9.28 | $128.89 \pm 2.65$ | $6.99 \pm 0.01$ | $124.17 \pm 3.12$ | $6.98 \pm 0.01$ | 0.18 | 0.20 |
| $5 * 10^{-2}$ | 11.05 | $152.95 \pm 3.92$ | $6.99 \pm 0.01$ | $116.96 \pm 2.69$ | $6.96 \pm 0.01$ | 0.00 | 0.00 |
| $1 * 10^{-1}$ | 14.22 | $185.26 \pm 4.53$ | $7.02 \pm 0.01$ | $119.03 \pm 2.63$ | $6.99 \pm 0.01$ | 0.00 | 0.00 |

Table 8: **Effect of $\epsilon$ tolerance in Algorithm 1, Line 19**: "Spec" is short for speculative, "Seq" is short for sequential, "PPL" is short for generative perplexity, "p-Val" is short for p-value from the t-test.

### F.6 LEFT-TO-RIGHT RESULTS

**Decoding Speed:** Our results extend to left-to-right prompting tasks. Firstly, the speculative decoding scheme still speeds up inference time without losing quality when evaluated with a prefix prompt and being asked to fill in the right. See Table 9.

| Method | Time (s) | NFE | Gen PPL | Entropy |
|--------|----------|-----|---------|---------|
| Sequential | $14.3 \pm 0.0$ | $383.0 \pm 0.0$ | $61.2 \pm 1.2$ | $7.35 \pm 0.02$ |
| Speculative | $13.3 \pm 0.1$ | $349.5 \pm 0.7$ | $60.0 \pm 1.1$ | $7.36 \pm 0.02$ |

Table 9: **Sequential versus Speculative Decoding in Left-to-Right Prompting:** Over 700 trials on WikiText (Merity et al., 2016), we prompt with a 128 token prefix, and have it complete the 384 token suffix.

Here is an example prompt for the left-to-right experiment (128 tokens):

```
<sep><cls> = Robert Boulter =<sep><cls><sep><cls>
Robert Boulter is an English film , television and theatre actor
. He had a guest @-@ starring role on the television series The
Bill in 2000 . This was followed by a starring role in the play
```

```
Herons written by Simon Stephens , which was performed in 2001 at
the Royal Court Theatre . He had a guest role in the television
series Judge John Deed in 2002 . In 2004 Boulter landed a role as
\" Craig \" in the episode \" Teddy 's Story \" of the television
series The Long Firm
```

Here is the continuation from Any-Subset Speculative Decoding (360 NFE to get 384 tokens):

```
 . In 2005's drug \"The Filipino Giant \" , he played Jake, a drug
addict who he meets at Hospital Montgomery , America's crooked
drug dealer (played by William Costolo) rented out by a local
dentist during Big Monkey Seasons. In 2006 Ted was a twin brother
as \" Bear\" and a full-blown hectic guy conducting scenic trips
with New Kingdom Railroad. In 2009 he played a middle-aged murder
victim in Common Core 2017 . Cast [ edit ]<sep><cls> Lawmakers
from both parties are leveraging the Lincoln-House dispute to
parrow votes below certain major milestones: Maryland Governor
cripples Democrat Ross RUMPRILLE, Md. \u2014 Maryland Gov. Larry
Hogan used a month and a half in the Republican-controlled
Senate to cut off a deadline for Senate voters to top out the
king's yacht. If the votes blow up at midnight on Friday, Hogan
told reporters, \"there's no deadline at all.\" On Thursday,
prosecutors in Montgomery said Hogan delayed issuing a state
pardon to Theo Roe, one of the nation's oldest pot thieves, as
he and Ashley Sandrow got significantly older. A state attorney
general's office in Baltimore clarified Wednesday that Hogan's
assurance that replicators will act quickly as they demand a
pardon said he looked forward to using increasing leverage
against his Republican challenger, including a deadline to avoid a
\u201cClosing Chance\u201d in the veto-proof cutoff. The attorney
general's office said Hogan anticipated the refusal to give a
pardon that would likely set up trades in the Deloitte Group and
other European and Asian investors who are sold by Washington and
other payments to lethal injections. \u201cIn order to circumvent
the blackout chance that moves the act out of the way if done
allowing them to protect Goodwill and funds from being put
```

**Left-to-Right Benchmarks:** Results on left-to-right benchmark tasks are in Table 10. On Lambada (Paperno et al., 2016), AS-ARM-PT performs the best out of all the models, supporting our claim that AS-ARMs are a powerful model class. SEDD-S (Gong et al., 2024) performs the worst. Something interesting we see is that for both DiffuGPT (which was finetuned from GPT) and AS-ARM-FT (which was finetuned from AS-ARM-PT's XLNet weights), performance on Lambada drops after finetuning. We hypothesize that the drop-off comes from the fact that the original models were already well-suited for the Lambada task. Lambada is a task to predict the last word (one or a few tokens) in a passage. GPT, with the left-to-right next token prediction objective, is well-suited for that. Similarly, AS-ARM-PT, being trained to infill 20% of a sequence, is well-suited to tasks with dense context. On the other hand, when one fine-tunes DiffuGPT and AS-ARM-FT, the desired output distribution shifts away from that. DiffuGPT now is trained to do any-order prediction (via CTMC diffusion process), and AS-ARM-FT shifts the focus to language generation from near-scratch (i.e., sparse context). Since models have finite capacity, it makes sense that the performance on the specialized last-word prediction task would drop. On HellaSwag (Zellers et al., 2019), AS-ARMs are a bit behind the baselines, but not extremely so. Overall, the results on sequential tasks suggest that our method is valid, even in settings different then it was intended for.

| Model | Lambda | HellaSwag |
|---|---|---|
| GPT-S (Gong et al., 2024) | 25.9 | 29.9 |
| DiffuGPT-S (Gong et al., 2024) | 21.6 | **33.4** |
| SEDD-S (Gong et al., 2024) | 12.4 | 30.2 |
| *AS-ARM-PT* | **26.6** | 28.0 |
| *AS-ARM-FT* | 14.1 | 28.3 |

Table 10: **Left-to-Right Benchmark Tasks:** Comparing AS-ARMs to baselines (both autoregressive and discrete diffusion) on left-to-right benchmark tasks.

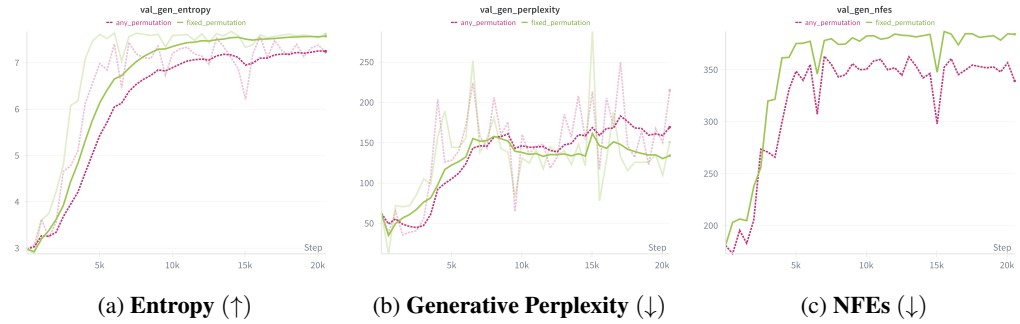

(a) **Entropy** ($\uparrow$)   (b) **Generative Perplexity** ($\downarrow$)   (c) **NFEs** ($\downarrow$)

Figure 3: **Fixed (Recursive Binary Lattice) Versus Any Permutation Mask Decomposition:** Validation loop metrics on generated sequences from each training strategy. The curves shown are for models trained with an effective batch size of 96 across four NVIDIA RTX A4000 devices. Each validation iteration has 36 sequences of length 512 tokens.

## G ABLATION

### G.1 MASK DECOMPOSITION ABLATION

Figure 3 shows the ablation of mask decomposition protocol described in Equation 4. The entropy (generation diversity) is consistently better when training with the recursive binary mask decomposition protocol ($2^N$ subsets, AS-ARM style) than with any of the $N!$ permutations (AO-ARM style). As training progresses, the generative perplexity is also better with the AS-ARM style. That being said, training on any of the $N!$ permutations (as opposed to restricting to the $2^N$ subsets) does result in fewer NFEs for ASSD generation, but this comes at the cost of lower quality outputs overall (as measured by the aforementioned entropy and generative perplexity).

### G.2 IMPACT OF MASKING DISTRIBUTION

Figure 4 shows the effect of the realization of the distribution of prompt (*i.e.*, masking) length from Section E.1. Since the validation task is to infill $95\%$ of a masked sequence, it is expected that training the model exclusively on shorter prompt lengths would be better than training the model on a mixture of long and short prompt lengths (which would dilute its capacity). Indeed, we see that, with respect to generative perplexity, the model trained with $m \sim \mathcal{U}[0.01, 0.10]$ outperforms the model trained with $m \sim \mathcal{U}[0.01, 0.85]$, where $m$ is the percentage of the source text that is given as prompt (*i.e.*, *un*masked). On entropy, the short prompt training strategy performs marginally better at low training steps, but the gap closes as training continues.

### G.3 IMPACT OF MASKING WARMUP

We also conduct a small-scale exploratory training run to test the sensitivity of our training scheme to hyperparameter choices, particularly the masking warmup in code generation (Section E.6) from StarCoder (Li et al., 2023). Due to computational constraints, we use batch size of 32 (4 per device, 4 accumulation steps, 2 A4000 GPUs). We have a final minimum masking rate of $85\%$ and a final maximum rate of $99\%$. One of the runs started from $15\%$ masking rate, and warmed up to the final

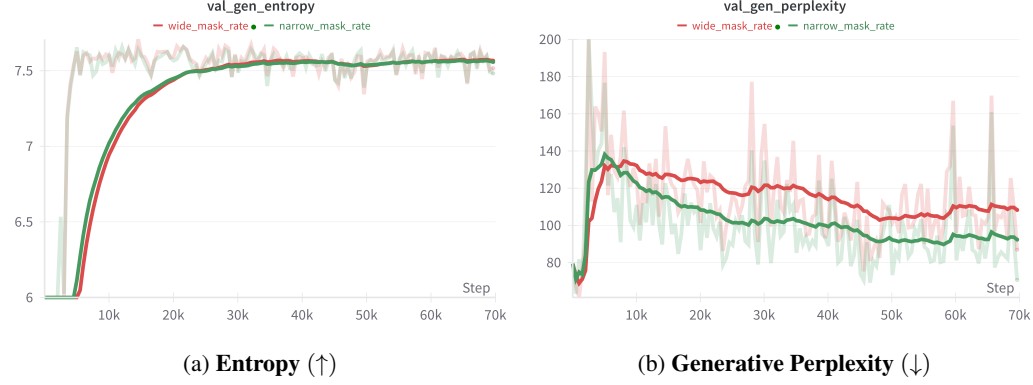

(a) **Entropy** ($\uparrow$)  (b) **Generative Perplexity** ($\downarrow$)

Figure 4: **Narrow ($1\% \rightarrow 10\%$) Versus Wide ($1\% \rightarrow 85\%$) Prompting Rates:** Validation loop metrics on generated sequences from each training strategy, as it relates to the distribution of prompt lengths in the train set. The curves shown are for models trained with an effective batch size of $320$ across five NVIDIA RTX 6000 Ada devices. Each validation iteration has $64$ sequences of length $512$ tokens from OpenWebText, where the task is to infill $95\%$ of the masked sequence given a $5\%$ prompt.

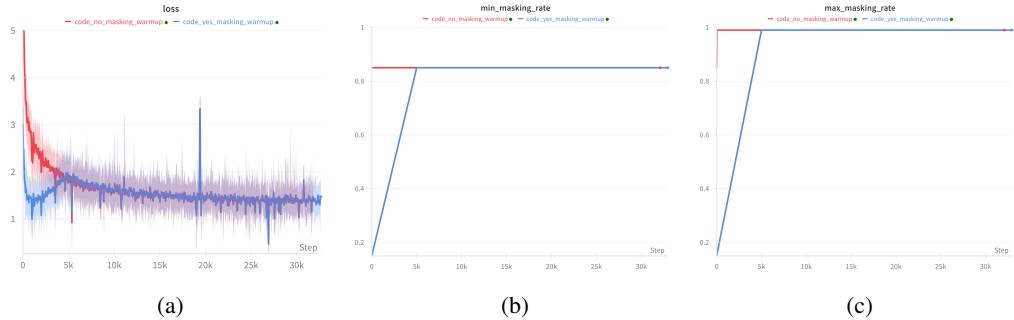

(a)  (b)  (c)

Figure 5: **Impact of Masking Warmup:** We compare code generation training loss with masking rate warmup and without masking rate warmup.

rates over 5000 steps, while the other run already started from $85\%$ masking rate and took only a single warmup step (meaning that it effectively trained at the final range since the beginning).

Looking at Figure 5, we see that at the beginning, the masking warmup makes a difference in the loss (since the task distribution is different); but after the warmup is finished, there is negligible difference. The conclusion is that the ultimate performance is fairly insensitive to the masking rate warmup.

### G.4 IMPACT OF PRETRAINED WEIGHTS

We also assess the impact of finetuning from the original XLNet pretrained weights (Yang, 2019) or training the architecture from scratch. Again, due to computational constraints, we set the batch size to 32 (4 per device, 4 accumulation steps, 2 A4000 GPUs). We have one run initialized from the pretrained weights, and the other initialized with random weights; the runs are otherwise identical. This experiment is done on OpenWebText (Gokaslan et al., 2019).

See Figure 6. We see that it is very important to initialize from the pretrained weights. The training loss and generative perplexity are very poor when initializing from random weights. That being said, we see the loss on the randomly initialized model improving, so we believe that given enough compute, it could eventually reach the desired performance.

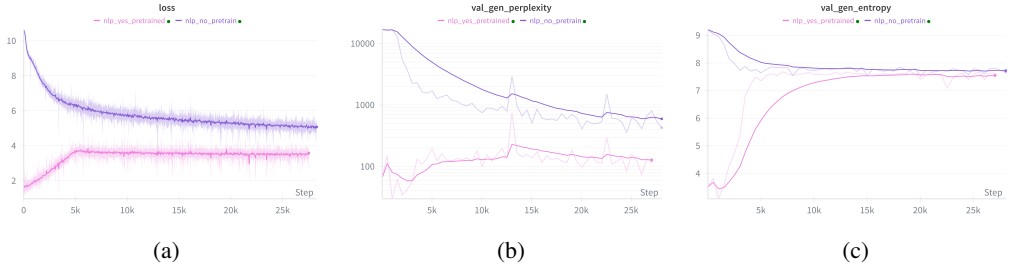

(a)  (b)  (c)

Figure 6: **Impact of Initialization:** We compare the performance of a model initialized with the XLNet pretrained weights (Yang, 2019) and a model initialized from scratch, both training on OpenWebText (Gokaslan et al., 2019).

## H    ADDITIONAL RELATED WORKS

### H.1    ANY-ORDER AUTOREGRESSIVE MODELS

There are some works from the pre-transformer era (Vaswani et al., 2017) that deal with the problem of arbitrary density estimation, namely NADE (Uria et al., 2014) and MADE (Germain et al., 2015). AO-ARMs also bear some resemblance to generative, transformer-based, masked image models (Chang et al., 2022; Du et al., 2024).

### H.2    DISCRETE DIFFUSION MODELS

There is also work on improving sampling. EDLM (Xu et al., 2024) learns a partition function to allow parallel sampling, while ensuring that the generated tokens adhere to a desired joint distribution. DDPD (Liu et al., 2024b) learns a planner network to optimize the step size taken with uniform diffusion models. JYS (Park et al., 2024) also works on optimizing the step schedule, although their method involves an optimization problem over (part of) the training dataset. (Zhao et al., 2024) work on predictor-corrector methods and introduce the $k-$Gillespie sampling algorithm. SDTT (Deschenaux & Gulcehre, 2024) distilled discrete diffusion models to achieve comparable generative performance with fewer sampling steps. However, (Zheng et al., 2024)'s work revealed a numerical precision error with discrete diffusion sampling that leads to unintentional low-temperature sampling.

### H.3    SPECULATIVE DECODING

Simple draft models include context-based and model-based n-grams (Stewart et al., 2024). Lookahead Decoding also relies upon parallel generation of n-grams, and does not need to train an additional draft model (Fu et al., 2024). Hydra (Ankner et al., 2024) and Medusa (Cai et al., 2024) augment the target language model with lightweight heads (which require additional training) to quickly predict additional tokens. The advantage of this approach, like ours, is that at least two tokens are guaranteed to be accepted on each loop iteration: the first token comes from the base model; if the next token is rejected, it is resampled from a combination of the proposal and oracle distributions. There are also works (Zhang et al., 2023) that, similar to ours, use the same model for drafting and verification; one such work is LayerSkip (Elhoushi et al., 2024). This is known as self-speculative decoding. A difference is that these methods skip model layers, while we skip inputs. There is also speculative diffusion decoding (Christopher et al., 2024), which uses a discrete diffusion model as the drafter for a larger autoregressive model. However, all these works are limited to the left-to-right regime, since the autoregressive oracle can only process $O(N)$ (all prefixes) prompting patterns - this is exponentially less than the $O(2^N)$ (all subsets) prompting patterns our method can handle.

The only work that we know of that uses a speculative decoding-like algorithm for non-left-to-right models is $\sigma$-GPT (Pannatier et al., 2024). However, their algorithm actually violates Theorem 1 and Theorem 2, meaning that it is not theoretically guaranteed to produce tokens from the correct target distribution, and can slow down sampling. See Appendix I for details.

## H.4 Constrained Decoding

In principle, constrained decoding techniques can be combined with vanilla AR models to accomplish infilling. Some recently popular approaches to constrained decoding model the constraint as a formal grammar (Geng et al., 2023; Beurer-Kellner et al., 2024) and only allow the left-to-right LLM to decode sequences that are valid under this grammar's parser. Other approaches use constrained beam search with trie data structures (De Cao et al., 2020) and Metropolis-Hastings Sampling (Miao et al., 2019). Recently, Ye et al. (2025) created an asymptotically unbiased constrained decoding algorithm based on importance sampling and GPU parallelism. However, while technically compatible with infilling, these works generally focus on other tasks, like entity retrieval or information extraction. Furthermore, constrained decoding incurs computational time overhead on top of the AR model, and biases the output distribution. In contrast, AS-ARMs natively handle arbitrary infilling tasks without overhead and directly estimate unbiased conditional probabilities.

## I COMPARISON TO SIGMA GPT

---

**Algorithm 3:** Token-based rejection sampling, reprinted from $\sigma$-GPT (Pannatier et al., 2024)

---

**Input:** $T$: minimum target length, $y$: any-order autoregressive model, $N_o$: number of orderings to sample, $\mathbb{X}$: prompt of length $t_0$

1  $t \leftarrow t_0$
2  **while** $t < T$ **do**
3  $\quad$ In parallel, compute distribution conditioned on prompt $p(x_i|\mathbb{X}), \forall i \in t, \dots, T$
4  $\quad$ In parallel, sample at every position $\tilde{x}_i \sim p(x_i|\mathbb{X}), \forall i \in t, \dots, T$
5  $\quad$ Draw $N_o$ random order $\sigma$ and in parallel, compute all logits $q\left(x_i|\mathbb{X}, \tilde{x}_{\sigma(<i)}\right), \forall i \in t, \dots, T$
6  $\quad$ In parallel sample $T - t$ variables $u_i \sim \mathcal{U}[0,1], \forall i \in t, \dots, T$ from a uniform distribution.
7  $\quad$ In parallel, compute the acceptance decision $a_i = u_i < \min\left(1, \frac{q\left(\tilde{x}_i|\mathbb{X}, \tilde{x}_{\sigma(<i)}\right)}{p(\tilde{x}_i|\mathbb{X})}\right)$ for every order.
8  $\quad$ Select the order that accepts the most tokens before seeing a first rejection.
9  $\quad$ Keep that order and add the $a$ accepted tokens before the first rejection to the prompt.
10 $\quad$ Set $t = t + a$
11 **end**

---

$\sigma$-GPT is a work that superficially seems to have a similar sampling algorithm as ours for any-order autoregressive models (Pannatier et al., 2024). However, as we will see, their algorithm (Algorithm 3) actually has subtle yet critical mistakes that lead to violations of Theorem 1 and Theorem 2, removing the theoretical guarantees.

### I.1 AMBIGUOUS FACTORIZATION ORDER AND JOINT DISTRIBUTION

Firstly, their algorithm does *not* provably give the correct joint distribution, because they randomly sample multiple factorization orders (Lines 5, 8) at each draft-accept cycle, given a prompt. This would re-introduce the consistency problem in Equation 3, *i.e.*, different orderings give different joint distributions. Thus, it is unclear what the true target distribution they are trying to match actually is, violating Theorem 2.

Another consequence of the multiple factorization orders sampled is that the number of NFEs could exceed the number of tokens eventually accepted, if the multi-order oracle evaluations all reject the speculated tokens. So, Theorem 1 is violated, which means that this scheme could potentially slow down sampling.

### I.2 LACK OF RESAMPLING STEP

Furthermore, their algorithm also does not have a resampling step in case of rejection. This decreases the number of tokens each iteration is guaranteed to accept from two to one (see Theorem 1's proof). This means that their algorithm is *not* mathematically guaranteed to reduce the number of function

evaluations, and can in theory increase it: in the case that only the first conditionally independent token from the draft is accepted, the function evaluation of the oracle did not lead to an extra token being accepted. This violates Theorem 1.

Furthermore, the lack of resampling potentially once again violates Theorem 2's guarantee that the returned distribution will match the joint. Using the notation and ideas from Appendix B, they have the "accept" (first) term in Equation 14, but they do not have the "reject" (second) term to balance the probability out. In fact, their Line 7 cannot even be considered to be proper rejection sampling, because it lacks the proper normalization for the decision threshold (Leviathan et al., 2023).

### I.3 Any-Order Speculative Decoding Addresses Problems

In contrast, by enforcing (Shih et al., 2022)'s recursive binary lattice mask decomposition, we ensure that given a prompt, there is *only one* correct path to calculating the joint conditional probability of the missing tokens. This makes it clear in our Algorithm 1 what the target distribution actually is, and our output provably matches that distribution (Theorem 2). Furthermore, we have resampling in Line 22. Combined with the single-path evaluation, this mathematically guarantees that we never increase the number of function evaluations above the number of masked tokens (Theorem 1).

## J    Absolute Speed Comparison

The primary goals of this paper are (1) to demonstrate that we can develop a lossless parallel decoding scheme for AO-ARMs/AS-ARMs; (2) to show that this long-forgotten model class can be competitive with the currently dominant model classes. Indeed, we achieve these goals.

That being said, it is also useful to understand how it compares to acceleration methods that are designed for other model classes. We compare against GPT-2-S (autoregressive) (Radford et al., 2019) and MDLM (discrete diffusion) (Sahoo et al., 2024). For GPT-2, we do regular decoding and KV caching. For MDLM, we vary the number of timesteps (*e.g.*, parallel inferences). For AS-ARMs, we do sequential decoding, ASSD with self-speculation, and ASSD with n-grams. We have each model generate $95\%$ of a 512-token sequence, prompted with the leftmost $5\%$ (taken from OpenWebText (Gokaslan et al., 2019)). We conduct 200 back-to-back trials per decoding method per model. Results are in Table 11.

| Steps | Gen PPL | Entropy | Time (s) |
|---|---|---|---|
| 512 | $96.5 \pm 3.5$ | $7.18 \pm 0.03$ | $5.88 \pm 0.01$ |
| 448 | $96.4 \pm 3.5$ | $7.21 \pm 0.03$ | $5.44 \pm 0.01$ |
| 384 | $101.8 \pm 3.9$ | $7.18 \pm 0.03$ | $4.94 \pm 0.01$ |
| 256 | $117.8 \pm 4.2$ | $7.32 \pm 0.02$ | $3.74 \pm 0.01$ |
| 128 | $137.9 \pm 4.1$ | $7.37 \pm 0.02$ | $2.08 \pm 0.00$ |

(a) MDLM (discrete diffusion) (Sahoo et al., 2024)

| KV Cache | Gen PPL | Entropy | Time (s) |
|---|---|---|---|
| No | $18.4 \pm 0.3$ | $7.04 \pm 0.02$ | $6.29 \pm 0.00$ |
| Yes | $18.2 \pm 0.3$ | $7.01 \pm 0.02$ | $2.92 \pm 0.00$ |

(b) GPT-2 (autoregressive) (Radford et al., 2019)

| Decoding Strategy | Gen PPL | Entropy | Model NFE | Aux NFE | Time (s) |
|---|---|---|---|---|---|
| Sequential | $65.5 \pm 3.1$ | $6.67 \pm 0.11$ | $487.0 \pm 0.0$ | $0.0 \pm 0.0$ | $19.24 \pm 0.11$ |
| ASSD (Self) | $65.1 \pm 3.9$ | $6.76 \pm 0.10$ | $435.8 \pm 3.2$ | $0.0 \pm 0.0$ | $17.13 \pm 0.13$ |
| ASSD (n-gram) | $63.2 \pm 3.1$ | $6.68 \pm 0.12$ | $359.6 \pm 6.4$ | $359.6 \pm 6.4$ | $15.09 \pm 0.28$ |

(c) AS-ARM

Table 11: **Absolute Speed Comparisons:** Speed and generative perplexity of decoding methods for various discrete generative model classes.

Before we get into the analysis, we want to give some caveats on the interpretation:

1. When comparing speeds, we should look at how one decoding scheme performs as compared to another decoding scheme for the *same* model – this provides an apples-to-apples comparison, as the only thing that changes is the decoding strategy. Otherwise, the architectural engineering, training strategy, etc., tint this comparison.

2. In that vein, it is somewhat flawed to compare generative perplexities for AS-ARMs with ASSD to the generative perplexities for GPT-2 and MDLM, because the tokenizer used for XLNet is different than that used by GPT-2 and MDLM. (XLNet has a 32,000 word vocabulary (Yang, 2019), while GPT-2 and MDLM use the same 50,257 word vocabulary (Radford et al., 2019).) Since length affects perplexity (and tokenization scheme affects length), generating 512 tokens from XLNet inherently is not comparable to generating 512 tokens from MDLM. Furthermore, because the generative perplexity judge is GPT-2 Large, one could imagine that it is biased towards outputs that were generated using the same GPT tokenization scheme that it uses. However, we could still discern rough trends from generative PPL comparisons. Inference time per token is a somewhat fairer comparison, if we wish to compare across models.

From this evaluation, our conclusions are:

1. Parallel decoding (up to a certain point), KV caching, and ASSD can all provide lossless acceleration for their respective model classes. KV caching on GPT-2 seems to be the fastest lossless accelerator at the moment. That being said, KV caching is orthogonal to ASSD, so future lossless accelerators could use some combination of both.

2. N-gram speculator can sometimes be empirically faster than self-speculative ASSD, due to the basically zero cost to go to an n-gram lookup table. That being said, n-gram does not have the theoretical guarantees of never decreasing function evaluations that the self-speculative variant does, so we should use it with caution as the problem difficulty scales.

3. MDLM and GPT-2 are currently faster than AS-ARM. This is likely related to the engineering optimizations that went into their architectural implementations, namely the use of Flash Attention (Dao et al., 2022). That being said, our approach, in principle, is flexible with respect to architecture.

4. GPT-2 has the best generative perplexity, presumably because it was specially designed for left-to-right tasks. That being said, it is not necessarily better on the other benchmark tasks in this paper, and generative perplexity is known to be a flawed metric that can be "hacked" (Zheng et al., 2024).

For future studies, we would have architectures that support training as an AS-ARM, diffusion language model, or regular autoregressive model, so that we can do a head-to-head comparison (with the same tokenizer, etc.). For this study, we did not emphasize that, as the point was to show that (1) lossless parallel generation is possible with AS-ARMs; (2) AS-ARMs are competitive with other model classes, and should be given more future research attention.

## K  OFF-THE-SHELF MODEL OUTPUTS

Text comes from the WikiText dataset (Merity et al., 2016). This is from the default XLNet model provided on Huggingface (*not* our finetuned), which was only trained to predict around 20% of tokens.

**Original Text**

<sep><cls> = Robert Boulter =<sep><cls><sep><cls> Robert Boulter is an English film , television and theatre actor . He had a guest @-@ starring role on the television series The Bill in 2000 . This was followed by a starring role in the play Herons written by Simon Stephens , which was performed in 2001 at the Royal Court Theatre . He had a guest role in the television series Judge John Deed in 2002 . In 2004 Boulter landed a role as " Craig " in the episode " Teddy 's Story " of the television series The Long Firm ; he starred alongside actors Mark Strong and Derek Jacobi . He was cast in the 2005 theatre productions of the Philip Ridley play Mercury Fur , which was performed at the Drum Theatre in Plymouth and the Menier Chocolate Factory in London . He was directed by John Tiffany and starred alongside Ben Whishaw , Shane Zaza , Harry Kent , Fraser Ayres , Sophie Stanton and Dominic Hall .<sep><cls> In 2006 , Boulter starred alongside Whishaw in the play Citizenship written by Mark Ravenhill . He appeared on a 2006 episode of the television series , Doctors , followed by a role in the 2007 theatre production of How to Curse directed by Josie Rourke . How to Curse was performed at Bush Theatre in the London Borough of Hammersmith and Fulham . Boulter starred in two films in 2008 , Daylight Robbery by filmmaker Paris Leonti , and Donkey Punch directed by Olly Blackburn . In May 2008 , Boulter made a guest appearance on a two @-@ part episode arc of the television series Waking the Dead , followed by an appearance on the television series Survivors in November 2008 . He had a recurring role in ten episodes of the television series Casualty in 2010 , as " Kieron Fletcher " . Boulter starred in the 2011 film Mercenaries directed by Paris Leonti .<sep><cls><sep><cls> = = Career = =<sep><cls><sep><cls><sep><cls> = = = 2000 – 2005 = = =<sep><cls><sep><cls> In 2000 Boulter had a guest @-@ starring role on the television series The Bill ; he portrayed " Scott Parry " in the episode , " In Safe Hands " . Boulter starred as " Scott

**Prompt**

_ _ _ _ _ _ _ _ _ _ _ _ _ _ _ _ _ _ _ _ _ _ _ _ _ _ _ _ _ _ _ _ _ _ _ _ _ the television_ _ _
_ _ _ _ _ _ _ _ _ _ _ in_ _ _ _ _ _ _ _ _ _ _,_ _ _ _ _ _ _ _ Court_ _ _ _ _ _ _ _ _ _ _ _ _ _ _ _ _
_ _ _ _ _ _ _ _ _ _ _ _ _ _ _ _ _ _ _ " _ _ _ _ _ _ _ _ _ _ _ _ _ _ _ _ _ _ _ _ and_ _ _ _ _
He_ _ _ _ 2005_ _ _ _ _ _ _ _ _ _ _ _ _ _ _ _ _ _ _ Plymouth_ _ _ _ _ _ _ _ _ _ _ _ _ _ _ _ _
_ _ _ _ _ _ _ _ _ _ _ _ _ _ _ _ _ _ _ _ _ _ Stanton and_ _ _ _ _ _ _ _ _ _ _ _ _ starred_ _ _ _ _
play Citizens_ _ _ _ _ _ _ _ _ _ _ a_ _ _ _ _ _ _ _ _ _ _ _ _ _ _ _ _ _ _ _ _ _ _ _ _ _ Josie_ _ _
_ _ _ _ _ _ _ _ _ _ _ _ _ _ _ _ _ _ _ _ _ _ ·_ _ _ _ _ _ _ _ _ _ _ _ _ _ _ _ _ _ _ _ _ _ _ key_ _ _ _
_ _ _ _ _ _ _ _ _ _ _ _ _ _ _ _ _ _ _ _ _ _ _ _ _ _ _ _ _ _ _ _ _ _ _ _ _ _ _ _ _ _ _ _ _ _ _ _ _ _
_ _ _ _ _ _ _ _ _ _ _ _ _ _ _ _ _ _ _ _ _ _ _ _ _ _ _ _ _ _ _ _ _ _ _ _ _ Merc_ _ _ _ _ _ _ _ _
_ _ _ _ _ _ _ _ _ =_ _ _ _ _ _ _ _ _ _ _ =_ _ _ _ _ _ _ _ _ _ _ _ _ _ _ _ _ _ _ _ _ _ _ _ _ _ _
the_ _ _ _ _ _ _ portrayed_ _ _ _ _ _ _ _ _ _ _ _ _ _ _ _ _ _ _ _ _ _ _ _ _ _ _ _ starred_ _ _ _ _

**Off-the-Shelf, Sequential Decoding**

300 300 300 300<eop> Ethnic or Ethnic Ethnically Ly Americans or Others or Others Beyond Ly Ly Ly Ly Ly Ly Ly Ly Ly Ly Ly Ly Ly Ly Ly Ly Ly Ly Ly Ly Ly Ly the television Ly Ly Ly Ly Ly Ly Ly Ly Ly Ly Ly Ly Ly in Ly Ly Ly Ly Ly Ly Ly North ), Ly Ly Ly Ly Ly Ly Ly Ly Court Ly Ly Ly Ly Ly Ly Ly Ly Ly Ly Ly Ly Ly Ly Ly Ly Ly Ly Ly Ly Ly Ly Ly Ly Ly Ly Ly Ly Ly Ly Ly Ly Ly Ly" Ly Ly Ly Ly Ly Ly Ly Ly Ly Ly Ly Ly Ly Ly Ly Ly Ly Ly Ly Ly Ly Ly and Ly Ly Ly Ly Ly He Ly Ly Ly Ly 2005 Ly Ly Ly Ly Ly Ly Ly Ly Ly Ly Ly Ly Ly Ly Ly Ly Ly Ly Ly Plymouth Ly Ly Ly Ly Ly Ly Ly Ly Ly Ly Ly Ly Ly Ly Ly Ly Ly Ly Ly Ly Ly Ly Ly Ly Ly Ly Ly Ly Ly Ly Ly Ly Ly Ly Ly Ly Stanton and Ly Ly Ly Ly Ly Ly Ly Ly Ly Ly Ly Ly starred Ly Ly Ly Ly Ly play Citizens Ly Ly Ly Ly Ly Ly Ly Ly Ly Ly a Ly Ly Ly Ly Ly Ly Ly Ly Ly Ly Ly Ly Ly Ly Ly Ly Ly Ly Ly Ly Ly Ly Ly Josie Ly Ly Ly Ly Ly Ly Ly Ly Ly Ly Ly Ly Ly Ly Ly Ly Ly Ly Ly Ly Ly Ly. Ly Ly Ly Ly Ly Ly Ly Ly Ly Ly Ly Ly Ly Ly Ly Ly Ly Ly Ly Ly Ly Lykey Ly Ly Ly Ly Ly Ly Ly Ly Ly Ly Ly Ly Ly Ly Ly Ly Ly Ly Ly Ly Ly Ly Ly Ly Ly Ly Ly Ly Ly Ly Ly Ly Ly Ly Ly Ly Ly Ly Ly Ly Ly Ly Ly Ly Ly Ly Ly Ly Ly Ly Ly Ly Ly Ly Ly Ly Ly Ly Ly Ly Ly Ly Ly Ly Ly Ly Ly Ly Ly Ly Ly Ly Ly Ly Ly Ly Ly Ly Ly Ly Ly Ly Merc Ly Ly Ly Ly Ly Ly Ly Ly Ly Ly Ly Ly Ly Ly Ly Ly Ly = Ly Ly Ly Ly Ly Ly Ly Ly Ly Ly = Ly Ly Ly Ly Ly Ly Ly Ly Ly Ly Ly Ly Ly Ly Ly Ly Ly Ly Ly Ly Ly Ly Ly Ly Ly the Ly Ly Ly Ly Ly Ly portrayed Ly Ly Ly Ly Ly Ly Ly Ly Ly Ly Ly Ly Ly Ly Ly Ly Ly; Ly Ly starred Ly Ly Ly Ly Ly

**Off-the-Shelf, Speculative Decoding (Algorithm 1)**

series and led starred starred starred starred starred starred starred starred starred starred starred starred starred starred starred starred starred starred starred starred starred starred starred starred starred starred starred starred starred starred starred starred starred starred and in the television major starred starred starred starred starred starred starred starred starred starred when the on in became being the title characters into scene as filming the, and too happen which came. The Screen Court Movie"""1996)( were cast they are booked cast to be cast moondicly on set as the movie upon movies too news faint as of taking pres "". They cast to appear on set as of other celebrities the her shee sheeee heee and and is casting as the Hee Hee she 2005 """"Uh Uh Uh Uh Uh Uh Uh, Plymouth listed on set the first here Hee hee she is casting as they are casting to appear as the scheduled here Hee hee he is cast as previously its scheduled her to appear as Stanton and into the Las Vegas during draft her seen her can be cast or starred as the backup for the play Citizens Las Vegas are casting ankle to appear assed or a the next scheduled Las Vegas Las ("""""" shortly before after the next scheduled Las Vegas Las Vegas Las "" Josie Las Vegas Las ""jinohlah,x Vegas Las Vegas Las Vegas Las Vegas Las Vegas Las Vegas "." Las Vegas Las Vegas Las Vegas Las Vegas Las Vegas Las Vegas Las Vegas Las Vegas Las Vegas Las Vegas Las Vegaskey Las Vegas Las Vegas Las Vegas Las Vegas Las Vegas Las Vegas Las Vegas Las Vegas Las Vegas Las Vegas Las Vegas Las Vegas Las Vegas Las Vegas Las Vegas Las Vegas Las Vegas Las Vegas Las Vegas Las Vegas Las Vegas Las Vegas Las Vegas Las Vegas Las Vegas Las Vegas Las Vegas Las Vegas Las Vegas Las Vegas Las Vegas Las Vegas Las Vegas Las Vegas Las Vegas Las Vegas Las Vegas Las Vegas Las Vegas Las Vegas Las Vegas Las Vegas Las Vegas Las Vegas Las Vegas Las Vegas Las Merc Las Vegas Las Vegas Las Vegas Las Vegas Las Vegas Las Vegas Las Vegas Las Vegas Las Vegas Las Vegas = Las Vegas Las Vegas Las Vegas Las Vegas Las Vegas Las = Las Vegas Las Vegas Las Vegas Las Vegas Las Vegas Las Vegas Las Vegas Las Vegas Las Vegas Las Vegas Las Vegas Las Vegas Las Vegas Las Vegas the Las Vegas Las Vegas Las Vegas Las portrayed Las Vegas Las Vegas Las Vegas Las Vegas Las Vegas Las Vegas Las Vegas Las Vegas Las Vegas Las Vegas Las Vegas Las Vegas starred Las Vegas Las Vegas Las

## L  FINETUNED MODEL OUTPUTS

Text comes from the WikiText dataset (Merity et al., 2016). Samples correspond to Table 1, with our model trained from $90\% \rightarrow 99\%$ masking rate, as described in Appendix E.2.

**Original Text**

<cls> AFI 's 10 Top 10 - # 6 Sports Film<sep><cls><sep><cls> = = Legacy = =<sep><cls><sep><cls> In the decades since its release , The Hustler has cemented its reputation as a classic . Roger Ebert , echoing earlier praise for the performances , direction , and cinematography and adding laurels for editor Dede Allen , cites the film as " one of those films where scenes have such psychic weight that they grow in our memories . " He further cites Fast Eddie Felson as one of " only a handful of movie characters so real that the audience refers to them as touchstones . " TV Guide calls the film a " dark stunner " offering " a grim world whose only bright spot is the top of the pool table , yet [ with ] characters [ who ] maintain a shabby nobility and grace . " The four leads are again lavishly praised for their performances and the film is summed up as " not to be missed . "<sep><cls> Paul Newman reprised his role as Fast Eddie Felson in the 1986 film The Color of Money , for which he won the Academy Award for Best Actor in a Leading Role . A number of observers and critics have suggested that this Oscar was in belated recognition for his performance in The Hustler . In 1997 , the Library of Congress selected The Hustler for preservation in the United States National Film Registry as " culturally , historically , or aesthetically significant . " Carroll and Rossen 's screenplay was selected by the Writers Guild of America in 2006 as the 96th best motion picture screenplay of all time . In June 2008 , AFI released its " Ten top Ten " — the best ten films in ten " classic " American film genres — after polling over 1 @,@ 500 people from the creative community . The Hustler was acknowledged as the sixth best film in the sports genre .<sep><cls> The Hustler is credited with sparking a resurgence in the popularity of pool in the United States , which had been on the decline for decades . The film also brought recognition to Willie Mosconi , who , despite having won multiple world championships , was virtually unknown to the general public . Perhaps the greatest beneficiary of the film 's popularity was a

**Prompt**

_ _ _ _ _ _ _ _ _ _ _ _ _ _ _ _ _ _ <sep>_ _ _ _ _ _ _ _ _ _ _ _ _ _ _ _ _ _ _ _ _,_ _ _ _ _ _ _
_ _ _ _ _ _ _ _ _ _ _ _ _ _ _ _ _ _ _ direction_ _ _ _ _ _ _ _ _ _ _ _ _ editor_ _ _ _ _ _ _ _ _
_ _ _ _ _ _ _ _ _ _ _ _ _ _ _ our_ _ _ _ _ _ _ _ _ _ _ _ _ _ _ _ _ _ _ _ _ _ _ _ _ _ _ _ _ _ _ _
_ _ _ _ _ _ _ _ _ _ _ _ _ _ _ _ _ _ _ _ _ _ _ _ _ _ _ _ _ _ _ _ the_ _ _ _ _ _ _ _ ]_ _ _ _
_ _ _ _ _ _ _ and_ _ _ _ _ _ _ _ _ _ lavish_ _ _ _ _ and_ film_ _ _ _ _ _ not_ _ _ _ _ _ _ _ _ _
_ _ _ _ _ role_ _ _ _ _ _ _ _ _ _ _ _ _ _ _ _ _ _ _ _ _ _ _ _ _ _ _ _ _ _ _ _ _ _ _ _ _ _ _ _ _
_ _ _ for_ _ _ _ _ _ _ _ _ _ _ _ Library_ _ _ _ _ _ _ _ _ _ _ _ _ _ _ _ _ _ _ _ _ _ _ _ _ _ _
_ _ _ _ _ _ _ _ _ _ _ _ _ _ _ _ _ _ _ _ _ _ _ _ _ _ _ _ _ _ _ _ _ _ _ FI_ _ _ _ _ _ Ten_ _
_ _ _ _ _ _ _ ten_ _ _ _ _ _ _ _ _ _ _ _ _ _ _ _ _ _ _ _ _ _ _ _ _ _ _ _ _ _ _ _ _ _ _ _ _ _
_ _ _ _ Hus_ _ _ _ _ _ _ _ _ _ _ _ _ _ _ _ _ _ _ _ _ _ _ _ _ _ _ _ _ also_ _ _ Willie_ _ _ _ _
_ _ _ _ _ _ _ _ _ _ _ _ _ _ _ _ _ _ _ _ _ _ _ _ _ _,_ _ _ _

**Finetuned, Sequential Decoding**

available to all user accounts. 3. Feel free to throw us a few questions and development suggestions.<sep><cls> GENEVA, Switzerland – The paper will not draft further stories, and we will, after all, ignore the party of journalistic freedom, which includes the right not to evaluate claims against anyone, except a dissenting direction from the scientific community. I assure you that we will continue the editor-in-chief role as a quarterly magazine where there will be no exemptions for writers of a particular viewpoint. This will lead our competition to host an active website, WMSU, which will host an automated competition about WMSU as well as to develop accredited methods of talking, researching and writing editorial articles in Swedish ("HF") and English (Roban Xismen in German). History [ edit ] Flawed motives for the art of the book have been debated much.[2] Until 2001, in ten of our 10 articles on 2001 and '02, we have used scientific forms, lavish rigor and style, and fake film. Since then, it has not happened to us. ARandom House, in cooperation with Wikipedia, consists in role-playing games and fantasy series literary novels with a unique approach to artistic media, including "Jeressande" by Audiom et les Hamerswein. See also [ edit ] Table of contents of articles for 2001-11 The intensive research effort of the Fabian Society and the Library and Museum of Great Britain on the RMA 1000 series includes more than nine dazzling articles, 50 "classics" and other pieces that garnered a mention from the Swedish press, an astute academic, sergeant d'Hoskin's Service, an item of art on the rise for WMSU in the 2002 World FI World Games.[3] Ten or more previous articles in February contributed to a ten-years archive by the L.B. Law Foundation of knowledge-urgent WB02 publications data. In 2005, Wercd et al. took the first step toward finishing "Moswart Huslas", which was published in the 1976 edition. In 2007, Munsry et al. repeated "White People's Rescue" also published events by Willie Miller and other Al Jazeera fresh and original voices. In a 2003 paper in the sociology of journalism, David Blabas reported, "The so called 'white men's

**Finetuned, Speculative Decoding (Algorithm 1)**

it learned to leave the visit." Read the folksy post—in full here...<sep><cls> CSI: Drayton, Part I Sir Ken & Honi Pratt Expedition Expeditions, Ltd Research & Development Canada U.S. History and Culture 1,200 Games History Remfacile Rerail will provide nine series of digital direction, redesign and multi-distribution services, leading original biographer and editor David Tatnow appears in the final stages of return service. "It's almost as fast as walking, getting the basics off our shelves," Chief Creative Officer Tony Paschus said. "We envy Historic Stadium, Ottawa and the story we can tell with access to our digital resource. The behind-the-scenes humblings cost the taxpayers more than $14 million last year and were the costliest part of CSI's entire qualification process isn't the only aspect of our financial problem [yet]. Players will have access to the most advanced titles, and collectible characters of any franchise. But they will be lavished with additional digital content and new film releases, in things that will not cut through Rui's new, toned down, strong-arm role—for example, when, in those spacesuits and used as rival generals, she can have dramatic conversations with everyone on her team. (Each character continues to be introduced during missions.) UK flags are symbolic of support for this transition from digital play to entertainment. Credit: Beaujorie Library Release date: 2015-01-25 Format: Remastered digital Download framerate: 57 kbps (a range of 1.2 to 54 kbps) Thumbnail up to 1920x1080 (16 in., 5 notches at 40.1°C) Planting Function: BFI 16 00 | Twelve Tens of catchests Producers: tenthreekorea-reports.gl.ca SUBSCRIBE TO Northeastern University's CSI: Drayton News & Games Tour: 51 Place (including Canada), Alberta (including Canada) Huston University University College of Art and Design MacAskill University of Waterloo (Note: Gamers 3 and under are welcome to sign up) See also: All About Willie Lewis<sep><cls> Posing for AC/DC's album's tour in October, D'Angelo Brakes of British Columbia proceeded with a classic 'DC' mixta

