# OpenReview forum: "Self-Speculative Decoding Accelerates Lossless Inference in Any-Order and Any-Subset Autoregressive Models"
_ICLR.cc/2026/Conference — ICLR 2026 Poster_

### Official Review · Reviewer_ummp · 2025-10-30

**Soundness:** 4
**Presentation:** 4
**Contribution:** 3
**Rating:** 8
**Confidence:** 4

**Summary:**

This paper proposes ASSD, a speculative decoding method for any-subset autoregressive models (AS-ARMs). ASSD yields about a 10% speedup over standard AS-ARM sampling and, thanks to its sound joint probabilistic modeling, outperforms much larger discrete diffusion models on code generation and infilling benchmarks.

**Strengths:**

1. The paper is very well written and easy to follow.

2. Unlike speculative decoding with drafting for ARMs, ASSD is guaranteed to *not* increase the number of calls, and does not need a separate drafting model.

3. Generation statistics for sequential decoding and ASSD are similar (Table 1), consistent with the theoretical guarantees (Lemma 1, Theorems 1 and 2).

4. I appreciate that the authors found and applied a fix to the DiffuLlama evaluation code and chose to compare against DiffuLlama with this correction, making the baseline stronger, instead of ignoring it.

**Weaknesses:**

1. A direct comparison of the speed and generation perplexity of ASSD against standard AR generation (e.g. GPT-2 with and without KV caching) and masked diffusion models (e.g. MDLM) is missing. I expect ASSD to be slower, but reporting these metrics is important for a complete evaluation, and does *not* diminish the value of the contributions, even as the observed ~10% speedup is notably lower than the 2-3x gains commonly reported for speculative decoding with a drafting model (e.g. in [1]).


2. Important details about hyperparameter choices are too often deferred to the appendix (e.g., lines 308-309 and 350-351). Since there is space available, a brief summary in the main text would be justified.

3. The motivation for the chosen training hyperparameters in code generation (appendix E7) is unclear. In particular, why was masking rate warmup used? Were there experiments comparing with and without it? Including results or rationale for this decision would strengthen the work and guide future research. Currenly, the choice appears somewhat ad-hoc.

4. Line 422: The "k" hyperparameter could be more clearly denoted as the number of speculated tokens.


[1] Fast Inference from Transformers via Speculative Decoding, Leviathan et al.

**Questions:**

1. On KV caching and causal masking: Lines 117-118 state that all prompt tokens attend to each other. If this is the case, KV caching should not be possible, as each new token would require recomputing previous activations due to bidirectional attention. Could the authors clarify how KV caching is handled in their experiments (appendix)? Is the prompt kept fixed during sampling, with new tokens using causal attention? Additionally, from line 213 ("density estimation via causal-like masking"), are the authors actually applying causal masking in the prompt?

2. The results in Table 3 show that performance on infilling multiple sequences in ROCStories is better than infilling a single sequence, which is counter-intuitive. I would expect the single-sequence case to be easier. Do the authors have an explanation for this? Additionally, after fine-tuning, the XLNet model performs worse at single-sentence infilling. Could this be related to the proportion of prompt tokens during the fine-tuning or something similar? This seems important to clarify.

3. Still, the speedup is much smaller than regular speculative decoding, do the authors have a sense of how to maybe push this further than 10% ?

4. I understand that no loss is applied to the prompt tokens during fine-tuning. Is that correct?

5. Have the authors considered training from scratch, compared to fine-tuning from an XLNet, even on some small synthetic datasets (if costs are prohibitive)? In any case, do you see some advantages or disadvantages of pre-training from scratch in terms of capabilities / inductive-biases of the final model?

---

> ### Author Response · Authors · 2025-11-15
> **Rebuttal Part 1**
>
> Thanks for your detailed review and insightful questions regarding our work. We’ve updated the manuscript, with changes in blue. We’ve considered your questions carefully, and here are our answers:
>
> # Weaknesses
> > **A direct comparison of the speed and generation perplexity of ASSD against standard AR generation (e.g. GPT-2 with and without KV caching) and masked diffusion models (e.g. MDLM) is missing.**
>
> Thanks for this suggestion. We report speed and generative perplexity for GPT-2, MDLM, and our AS-ARM. We have each model generate 95% of a 512-token sequence, prompted with the leftmost 5% (taken from OpenWebText). Reported results are for 200 back-to-back trials per decoding method per model. For GPT-2, we do regular decoding and KV-cached decoding. For MDLM, we vary the number of timesteps (aka parallel decoding), and for AS-ARM, we do regular decoding and ASSD (with self-speculation and n-gram speculator).  The following tables report mean and SEM.
>
> **MDLM:**
> | Steps | Gen PPL | Entropy | Time (s) |
> | - | - | - | - |
> | 512 | 96.5 +/- 3.5 | 7.18 +/- 0.03 | 5.88 +/- 0.01 |
> | 448 | 96.4 +/- 3.5 | 7.21 +/- 0.03 | 5.44 +/- 0.01 |
> | 384 | 101.8 +/- 3.9 | 7.18 +/- 0.03 | 4.94 +/- 0.01 |
> | 256 | 117.8 +/- 4.2 | 7.32 +/- 0.02 | 3.74 +/- 0.01 |
> | 128 | 137.9 +/- 4.1 | 7.37 +/- 0.02 | 2.08 +/- 0.00 |
>
> **GPT-2:**
> | KV Cache | Gen PPL | Entropy | Time (s) |
> | - | - | - | - |
> | No | 18.4 +/- 0.3 | 7.04 +/- 0.02 | 6.29 +/- 0.00 |
> | Yes | 18.2 +/- 0.3 | 7.01 +/- 0.02 | 2.92 +/- 0.00 |
>
> **AS-ARM:**
> | Decoding Strategy | Gen PPL | Entropy | Model NFE | Aux NFE | Time (s) |
> | - | - | - | - | - | - |
> | Sequential | 65.5 +/- 3.1 | 6.67 +/- 0.11 | 487.0 +/- 0.0 | 0.0 +/- 0.0 | 19.24 +/- 0.11 |
> | ASSD (Self) | 65.1 +/- 3.9 | 6.76 +/- 0.10 | 435.8 +/- 3.2 | 0.0 +/- 0.0 | 17.13 +/- 0.13 |
> | ASSD (n-gram) | 63.2 +/- 3.1 | 6.68 +/- 0.12 | 359.6 +/- 6.4 | 359.6 +/- 6.4 | 15.09 +/- 0.28 |
>
> Before the analysis, we give some caveats on interpretation:
> 1. When comparing results, we should look at how one decoding scheme performs as compared to another decoding scheme for the *same* model - this provides an apples-to-apples comparison, as the only thing that changes is the decoding strategy. Otherwise, the architectural engineering, training strategy, etc., tint this comparison.
> 2. In that vein, it is somewhat flawed to compare generative perplexities for AS-ARMs with ASSD to the generative perplexities for GPT and MDLM, because XLNet has its own 32,000 vocabulary tokenizer, while GPT and MDLM use the same 50,257 vocabulary tokenizer. Since length affects perplexity (and tokenization scheme affects length), generating 512 tokens from XLNet inherently is not comparable to generating 512 tokens from MDLM. Furthermore, the perplexity judge is GPT-2 Large, so it may be biased towards outputs that were generated using the same GPT tokenization scheme that it uses. That being said, we could still discern rough trends from generative PPL comparisons, albeit with a huge grain of salt. Inference time per token is also a somewhat fairer comparison.
>
> From this evaluation, our conclusions are:
> 1. Parallel decoding (up to a certain point), KV caching, and ASSD can all provide lossless acceleration for their respective model classes. KV caching on GPT-2 currently seems to be the fastest lossless accelerator. *KV caching is orthogonal to ASSD*, so future lossless acceleration strategies could combine them.
> 2. N-gram speculator can sometimes be empirically faster than self-speculative ASSD, due to the basically zero cost to go to an n-gram lookup table: indeed, we see a **20% speedup** over sequential here. That being said, n-gram does not have the theoretical guarantees of never decreasing NFEs that the self-speculative variant does, so we should use it with caution as problem difficulty scales.
> 3. MDLM and GPT-2 are currently faster than AS-ARM. This is likely related to the engineering optimizations that went into their architectural implementations, namely the use of Flash Attention [4]. *As you said, this doesn't diminish our core contributions of developing lossless acceleration algorithms for AS-ARMs and showing AS-ARMs' competitiveness.* We note that in principle, our approach is architecture-flexible.
> 4. GPT-2 has the best generative perplexity, presumably because it was specially designed for left-to-right tasks. That being said, it is not necessarily better on the other benchmark tasks in this paper, and generative perplexity is known to be a flawed metric that can be “hacked” [3].
>
> For future studies, we would have architectures that support training as an AS-ARM, diffusion language model, or regular autoregressive model, so that we can do a head-to-head comparison (with the same tokenizer, etc.). For this study, we did not emphasize that, as the point was to show that (1) lossless parallel generation is possible with AS-ARMs; (2) AS-ARMs are competitive with other model classes, and should be given more future research attention.

---

> ### Author Response · Authors · 2025-11-15
> **Rebuttal Part 2**
>
> **Important details about hyperparameter choices are too often deferred to the appendix (e.g., lines 308-309 and 350-351). Since there is space available, a brief summary in the main text would be justified.**
>
> We have more hyperparameter details in the *updated Section 6 (in blue).*
>
> **The motivation for the chosen training hyperparameters in code generation (appendix E7) is unclear. In particular, why was masking rate warmup used? Were there experiments comparing with and without it?**
>
> Thanks for inspiring us to reconsider this. We conduct a new ablation study with and without masking rate warmup. Although the loss differs at the beginning (while the masking rate is warming up), after it has warmed up, the loss basically does not differ. So, we could go with or without masking rate warmup. (As such, we do not retrain the model, since we don’t have enough compute at the moment.) This is good, because it shows that our training scheme is robust to hyperparameter choice. *See Figure 5 and Section G.3 in the new manuscript.*
>
> **Line 422: The "k" hyperparameter could be more clearly denoted as the number of speculated tokens.**
>
> Thanks. We clarify this now.
>
> # Questions
>
> **On KV caching and causal masking: Lines 117-118 state that all prompt tokens attend to each other. If this is the case, KV caching should not be possible, as each new token would require recomputing previous activations due to bidirectional attention. Could the authors clarify how KV caching is handled in their experiments (appendix)? Is the prompt kept fixed during sampling, with new tokens using causal attention?**
>
> Yes, the prompt is kept fixed during sampling, with new tokens using causal attention. We can still cache the content embeddings for the new generated tokens, but they have causal attention, so we don’t have to keep recomputing the prompt.
>
> **Additionally, from line 213 ("density estimation via causal-like masking"), are the authors actually applying causal masking in the prompt?**
>
> No. Since we only care to evaluate the likelihood of completions, we don’t apply causal masking on the prompt, but it would not be incorrect to do so.
>
> **The results in Table 3 show that performance on infilling multiple sequences in ROCStories is better than infilling a single sequence, which is counter-intuitive. I would expect the single-sequence case to be easier. Do the authors have an explanation for this? Additionally, after fine-tuning, the XLNet model performs worse at single-sentence infilling. Could this be related to the proportion of prompt tokens during the fine-tuning or something similar? This seems important to clarify.**
>
> It seems that you have answered your own question. :) This is certainly related to the proportion of prompt tokens during the fine-tuning. When we finetune, we ultimately train the model to generate almost from scratch, seeing only between 1% to 10% of the total sequence length as prompt. As such, the model is better-suited for tasks with smaller prompts, i.e., infilling 3 out of 5 sentences, as opposed to infilling 1 out of 5.
>
> Indeed, there does seem to be some sort of forgetting phenomenon. The original XLNet was trained to infill about 20% of a sequence [1], which would correspond to the single-sentence infilling task. That’s why the original XLNet (AS-ARM-PT) did better on infilling 1 out of 5 than AS-ARM-FT did.
>
> Overall, this shows that even if a task (e.g., single sentence infilling) is usually easier than another task (e.g., multi-sentence infilling), the training distribution of AS-ARMs can make a huge performance difference, something that is also implied by Shih et al [2].

---

> ### Author Response · Authors · 2025-11-15
> **Rebuttal Part 3**
>
> **Still, the speedup is much smaller than regular speculative decoding, do the authors have a sense of how to maybe push this further than 10% ?**
>
> Firstly, at least on the new left-to-right generation experiments (see Rebuttal Part 1), it seems that using alternative speculators like n-grams could *push the speedup to around 20%* (although this is empirically observed and not theoretically guaranteed).
>
> Another way is to do AO-ARM-style as opposed to AS-ARM style training; that is, we allow training on all $N!$ permutations, rather than all $2^N$ subsets (see Section 2). The reason why this could help is because the model would have more practice “skipping” over positions when decoding, which could lead to better speculation.
>
> The reason why we didn’t do this is because it hurts the learning of the joint distribution by diluting the finite model capacity over even more tasks. This means that although the speculation might be more likely to adhere to the learned joint distribution, the learned joint distribution itself would be less reliable, which means that output quality is ultimately worse. *See our updated Figure 3 for the ablation curves.*
>
> We think another way to get above 10% boost is to extend this technique to billion-parameter models (which we did not have resources to train in this study). For such large models, there are a lot of options of cheaper speculators to use to shave off time. In contrast, with small models like the ones we used, there are few options for cheaper models besides self-speculation and n-grams, and overhead time could make a difference too.
>
> **I understand that no loss is applied to the prompt tokens during fine-tuning. Is that correct?**
>
> This is correct.
>
> **Have the authors considered training from scratch, compared to fine-tuning from an XLNet, even on some small synthetic datasets (if costs are prohibitive)? In any case, do you see some advantages or disadvantages of pre-training from scratch in terms of capabilities / inductive-biases of the final model?**
>
> That’s a really good question. We pulled together some spare compute and did a quick training run on OpenWebText, initializing from a pretrained XLNet, and initializing from scratch. Basically, while we do see a slow decrease in loss for the initialization from scratch, the loss and validation metrics are far worse than we would get from initializing from a pretrained XLNet. Given more compute, it could be feasible to train from scratch, but in the compute-limited regime, finetuning from a pretrained XLNet is the best move. *See Figure 6 and Section G.4 in the new manuscript.*
>
> ~
>
> [1] Yang, Zhilin, et al. "Xlnet: Generalized autoregressive pretraining for language understanding." Advances in neural information processing systems 32 (2019).
>
> [2] Shih, Andy, Dorsa Sadigh, and Stefano Ermon. "Training and inference on any-order autoregressive models the right way." Advances in Neural Information Processing Systems 35 (2022): 2762-2775.
>
> [3] Zheng, Kaiwen, et al. "Masked diffusion models are secretly time-agnostic masked models and exploit inaccurate categorical sampling." ICLR 2025.
>
> [4] Dao, Tri, et al. "Flashattention: Fast and memory-efficient exact attention with io-awareness." Advances in neural information processing systems 35 (2022): 16344-16359.
>
> ~
>
> **Thanks for your review, and please let us know if you have any further questions!**

---

### Official Review · Reviewer_BarN · 2025-11-01

**Soundness:** 3
**Presentation:** 3
**Contribution:** 3
**Rating:** 4
**Confidence:** 3

**Summary:**

The paper introduces a new algorithm called Any-Subset Speculative Decoding (ASSD) to significantly speed up text generation for a special class of models called Any-Subset Autoregressive Models (AS-ARMs). This method allows for parallel generation of multiple tokens at once without any loss in output quality, effectively solving a major bottleneck in generating text for tasks like infilling (filling in missing parts of a text).

**Strengths:**

Lossless Generation: It is mathematically proven to produce text from the exact same distribution as slow, one-by-one sequential decoding. There is no quality degradation.
No Slowdown: The paper proves that the number of model calls will never exceed the number of tokens being generated. This is a critical guarantee that standard speculative decoding lacks, as a poor draft model can actually slow things down.
Free Draft Model: Since the AS-ARM acts as its own drafter and verifier, there is no need to train or maintain a separate, smaller draft model, saving on resources.
Enhanced Versatility: The method is designed for "any-subset" tasks, meaning it can handle an exponential number of infilling patterns (O(2^N)) compared to the linear number of prefix-based patterns (O(N)) handled by standard autoregressive models.

**Weaknesses:**

1. Scope beyond infilling use cases is not defined or compared.
2. Detailed analysis of accuracy of these models is not mentioned.

**Questions:**

1. How do AO-ARMs compare with regular ARMs and do you think modeling the joint probability distribution in AO-ARMs facilitate the speculation intrinsically while compromising on model quality as compared to regular ARMs?

---

> ### Author Response · Authors · 2025-11-13
> **Rebuttal Part 1**
>
> Thanks for recognizing the theoretical and methodological contributions of our work! Here are our answers to your concerns:
>
> # Weaknesses
> **Scope beyond infilling use cases is not defined or compared.**
>
> We think you are wondering about the performance on left-to-right text generation tasks. We actually did address this, but due to space limitations, the results were put in the appendix. For your convenience, we reprint them here, with some commentary:
> | Method | Time (s) | NFE | Generative Perplexity | Entropy |
> | --- | --- | --- | --- | --- |
> | Sequential | 14.3 +/- 0.0 | 383.0 +/- 0.0 | 61.2 +/- 1.2 | 7.35 +/- 0.02 |
> | Speculative | 13.3 +/- 0.1 | 349.5 +/- 0.7 | 60.0 +/- 1.1 | 7.36 +/- 0.02 |
>
> Over 700 trials on WikiText, we prompt our AS-ARM with a 128 token prefix, and have it predict the next 384 tokens.
> We see that our Any-Subset Speculative Decoding, even on this non-infilling task, still provides the speedup over sequential (aka one-by-one decoding) without drop in quality, as predicted by our theory and shown in the infilling experiments. We also realize that the appendix doesn’t contain text samples, so we print them here (they will be added to the PDF shortly):
>
> Prompt (128 tokens):
> ```
> <sep><cls></s> = Robert Boulter =<sep><cls></s><sep><cls></s> Robert Boulter is an English film , television and theatre actor . He had a guest @-@ starring role on the television series The Bill in 2000 . This was followed by a starring role in the play Herons written by Simon Stephens , which was performed in 2001 at the Royal Court Theatre . He had a guest role in the television series Judge John Deed in 2002 . In 2004 Boulter landed a role as \" Craig \" in the episode \" Teddy 's Story \" of the television series The Long Firm
> ```
>
> Continuation (360 NFE to get 384 tokens):
> ```
>  . In 2005's drug \"The Filipino Giant \" , he played Jake, a drug addict who he meets at Hospital Montgomery , America's crooked drug dealer (played by William Costolo) rented out by a local dentist during Big Monkey Seasons. In 2006 Ted was a twin brother as \" Bear\" and a full-blown hectic guy conducting scenic trips with New Kingdom Railroad. In 2009 he played a middle-aged murder victim in Common Core 2017 . Cast [ edit ]<sep><cls></s> Lawmakers from both parties are leveraging the Lincoln-House dispute to parrow votes below certain major milestones: Maryland Governor cripples Democrat Ross RUMPRILLE, Md. \u2014 Maryland Gov. Larry Hogan used a month and a half in the Republican-controlled Senate to cut off a deadline for Senate voters to top out the king's yacht. If the votes blow up at midnight on Friday, Hogan told reporters, \"there's no deadline at all.\" On Thursday, prosecutors in Montgomery said Hogan delayed issuing a state pardon to Theo Roe, one of the nation's oldest pot thieves, as he and Ashley Sandrow got significantly older. A state attorney general's office in Baltimore clarified Wednesday that Hogan's assurance that replicators will act quickly as they demand a pardon said he looked forward to using increasing leverage against his Republican challenger, including a deadline to avoid a \u201cClosing Chance\u201d in the veto-proof cutoff. The attorney general's office said Hogan anticipated the refusal to give a pardon that would likely set up trades in the Deloitte Group and other European and Asian investors who are sold by Washington and other payments to lethal injections. \u201cIn order to circumvent the blackout chance that moves the act out of the way if done allowing them to protect Goodwill and funds from being put
> ```
>
> Additionally, we also try some non-infilling benchmark tasks, namely Lambada [1] and HellaSwag [2]. Lambada is a last-word prediction task, while HellaSwag is a multiple choice answering task.
>
> | Model | Lambada | HellaSwag |
> | --- | --- | --- |
> | GPT-S | 25.9 | 29.9 |
> | DiffuGPT-S | 21.6 | **33.4** |
> | SEDD-S | 12.4 | 30.2 |
> | *AS-ARM-PT* | **26.6** | 28.0 |
> | *AS-ARM-FT* | 14.1 | 28.3 |
>
> On Lambada, AS-ARM-PT (pretrained on dense context) [3] performs the best out of all the models, supporting our claim that AS-ARMs are a capable model class. On HellaSwag, AS-ARMs are marginally behind the baselines, but not extremely so. Overall, the results on sequential benchmark tasks suggest that AS-ARMs are powerful and versatile, being competitive even on tasks they were not intended for. More details can be found in Appendix F6 (“Left-to-Right Results”).

---

> ### Author Response · Authors · 2025-11-13
> **Rebuttal Part 2**
>
> **Detailed analysis of accuracy of these models is not mentioned.**
>
> In regards to “accuracy”, we think you may be asking one of two things, since the notion of “accuracy” metrics is not well-defined for generative NLP (as opposed to ImageNet classification in computer vision): (1) Does our proposed Any-Subset Speculative Decoding method maintain the quality of the outputs? (2) How good are the AS-ARM models as compared to other models?
>
> In regards to (1), our proposed Any-Subset Speculative Decoding is theoretically guaranteed to give outputs from the same distribution as the learned sequential distribution (Theorem 2), and the empirical results corroborate this (Table 1). In this sense, the *“accuracy” is maintained with our method*.
>
> In regards to (2), our models compare favorably in performance to a plethora of other models, including autoregressive GPT models, discrete diffusion models (MDLM, SEDD), and even finetuned LLaMA models that are 50 times larger. Since *accuracy is not a typical metric in generative NLP, we evaluate on multiple standard benchmarks*, including: HumanEval Infilling (code generation), ROCStories (context-based story completion), Lambada (sentence completion), HellaSwag (multiple-choice reading comprehension). On these evaluations, AS-ARMs are **generally on par with or exceeding the competing paradigms** (see Table 2, 3, 10). In this sense, **our method is “as accurate” as, if not more so than, other methods**.
>
> # Questions
>
> **How do AO-ARMs compare with regular ARMs?**
>
> From a design perspective, AO-ARMs/AS-ARMs can handle all the tasks regular ARMs can ($O(N)$ prefixes), in addition to $O(2^N)$ infilling patterns. As a concrete example, take the prompt:
> ```The President is [MASK] [MASK].```
> Both AO-ARMs and ARMs can handle this prompt, and complete it as:
> ```The President is Donald Trump.```
>
> But, with the prompt:
> ```[MASK] President [MASK] [MASK] Obama```,
> *only AO-ARMs/AS-ARMs can complete this prompt* as:
> ```The President was Barack Obama```,
> while *ARMs (due to their left-to-right design) cannot* complete this prompt.
>
> **Do you think modeling the joint probability distribution in AO-ARMs facilitate the speculation intrinsically while compromising on model quality as compared to regular ARMs?**
>
> We indeed believe that modeling multiple factorizations of the joint probability distribution in AO-ARMs facilitates speculation intrinsically. The model is trained to generate in *arbitrary orders* and integrate *context scattered in noncontiguous locations*. These capabilities facilitate parallel generation, which can be thought of as generating in an *order* where we *“skip” context from some locations*.
>
> With regards to model quality, we do not see substantial compromise in empirical performance as compared to regular ARMs. On benchmark tasks, AO-ARMs/AS-ARMs are still competitive with and/or beating the comparably sized GPT2 autoregressive models (see first part of this rebuttal, and Tables 3,10 in main text).
>
> That being said, Shih et al do provide evidence that training difficulty does increase with the number of joint distribution factorizations (i.e., token orderings) the model has to learn [4]. (This phenomena was also observed in XLNet [3].) Technically, an autoregressive model is a special case of an AO-ARM, where it only has to learn one ordering, which would suggest that ARMs are easier to train, as they have exponentially fewer tasks to address. But, our experiments suggest that *a proper training recipe can close the gap to a great extent*.
>
> [1] Paperno, Denis, et al. "The LAMBADA dataset: Word prediction requiring a broad discourse context." Proceedings of the 54th annual meeting of the association for computational linguistics (volume 1: Long papers). 2016.
>
> [2] Zellers, Rowan, et al. "Hellaswag: Can a machine really finish your sentence?." arXiv preprint arXiv:1905.07830 (2019).
>
> [3] Yang, Zhilin, et al. "Xlnet: Generalized autoregressive pretraining for language understanding." Advances in neural information processing systems 32 (2019).
>
> [4] Shih, Andy, Dorsa Sadigh, and Stefano Ermon. "Training and inference on any-order autoregressive models the right way." Advances in Neural Information Processing Systems 35 (2022): 2762-2775.
>
> **Thanks for your reviewing effort, and we hope you will consider raising your score! Please let us know if you have any further questions.**

---

### Official Review · Reviewer_yrNB · 2025-11-02

**Soundness:** 3
**Presentation:** 3
**Contribution:** 2
**Rating:** 8
**Confidence:** 3

**Summary:**

This paper proposes to combine any-order models with speculative decoding. The method approaches any-order modeling by sampling a mask then using an encoder-decoder architecture to encode all previous tokens and produces remaining tokens autoregressively from left-to-right. The generation process is sped-up via self-spec-decoding, which uses parallel sampling (as in masked diffusion models) to draft and autoregressive mode to verify.

Empirical results on infilling in WikiText, coding, and ROCstories demonstrate a reasonable accuracy-speed tradeoff.

**Strengths:**

The method is reasonable, and exploring speculative decoding for infilling is novel. The writing was also easy to understand.

**Weaknesses:**

The main weakness is that the speedups are relatively small, and the acceptance length of 2.24 tokens per draft (Section 7.1) seems quite small. It's possible this is an artifact of model size. The evaluations are constrained by computational resources, so I will not let this affect my review too strongly. Additionally, the evaluations are a bit outdated as well. Again, this seems to be a limitation of computational resources.

**Questions:**

The token comparison in Table 3 does not seem to be fair, as I believe those are MDLM pretraining tokens.

---

> ### Author Response · Authors · 2025-11-13
> **Response**
>
> Thanks for recognizing the novelty of our work and your positive assessment! To address your concerns:
>
> # Weaknesses:
>
> **The main weakness is that the speedups are relatively small:**
>
> It’s important to remember that **these speedups are basically a “free lunch”**: we can get the ~10% speedup without sacrificing any quality. Indeed, we emphasize that our method is theoretically guaranteed to *never increase* the number of function evaluations (Theorem 1) and *never degrade* quality (Theorem 2). We contrast this to discrete diffusion models, which could provide speedups, but with no guarantee on the quality of the outputs (which often declines as function evaluations are decreased) [1]. We will update the PDF with this clarification shortly.
>
> **Again, this seems to be a limitation of computational resources.**
>
> We also appreciate your understanding of our lack of computational resources: as you know, we researchers face a perpetual battle to scrap together enough compute to realize our scientific visions :)
>
> # Questions:
>
> **The token comparison in Table 3 does not seem to be fair, as I believe those are MDLM pretraining tokens.**
>
> We looked at Table 3 as well as the MDLM paper [2] again, and are not sure what you mean by MDLM pretraining tokens: to our knowledge, MDLM has only one training phase. Please let us know if this answers your question, or if you had something else in mind.
>
> *Looking forward to any additional feedback we can address for you!*
>
> [1] Lou, Aaron, Chenlin Meng, and Stefano Ermon. "Discrete diffusion modeling by estimating the ratios of the data distribution." ICML 2024.
>
> [2] Sahoo, Subham, et al. "Simple and effective masked diffusion language models." Advances in Neural Information Processing Systems 37 (2024): 130136-130184.

---

> > ### Comment · Reviewer_yrNB · 2025-11-27
> >
> > On table 3: MDLM only has one training phase, whereas AS-ARM models are fine-tuned from XLNet models. As such, it's not completely fair to compare the MDLM single-phase tokens to AS-ARM's second phase tokens, not including the XLNet's training tokens. Unless of course the 33B and 12B numbers include the number of tokens XLNet was trained on. This is a pretty minor detail, though.

---

> ### Author Response · Authors · 2025-11-27
> **Reply to Reply**
>
> Thanks for the clarification.
>
> # Clarified Presentation
> (1) We change the presentation of Table 3 to indicate when we add *additional* finetuning tokens after pretraining. The 12B tokens for AS-ARM-FT are *after* pretraining. See new manuscript.
>
> # Vagueness in XLNet Pretraining Details
> (2) In the course of considering your question, we investigated further, and found out that the number of pretraining tokens for XLNet-Base (which we call AS-ARM-PT: pretrained) seems to have not been publicly released by the authors [1].
>
> They did release the size of their dataset, which was 33B tokens. We previously thought they used a single-epoch scheme, which is why we originally reported that they used 33B pretraining tokens.
>
> The best information we could find was their Table 7 (pg 13) and Section 3.1 (pg 6), which reports the training hyperparameters for their 340M parameter XLNet-Large model: this would amount to two trillion tokens [1]. However, we are dealing with the 110M XLNet-Base model. Checking their GitHub, they say that XLNet-Base is "trained on full data (different from the one in the paper)". It's a bit unclear what this means, but we guess that they are trying to say that they use the same training recipe as XLNet-Large.
>
> Accordingly, we update the manuscript with this information.
>
> # Still a Fair Comparison
> (3) Even with the (potentially) large pretraining corpus, we think this is still a reasonable comparison, because we benchmark against the DiffuGPT baseline [3], which is initialized from a pretrained GPT-2. Indeed, the GPT-2 initialization also does not report how many tokens/epochs they trained for [4], but we speculate that the OpenAI team committed many resources to their flagship model.
>
> Furthermore, as shown by DiffuGPT and DiffuLLaMA [3], we think it's actually a strength that our framework can leverage off-the-shelf pretrained models and conduct light finetuning (which uses much fewer tokens than training MDLM [5] or SEDD [6] from scratch) to achieve target tasks of interest. This could provide a more cost-effective method than having to train a model from scratch.
>
> ~
>
> Thanks for your positive assessment, and let us know if you have further questions!
>
> [1] Yang, Zhilin, et al. "Xlnet: Generalized autoregressive pretraining for language understanding." Advances in neural information processing systems 32 (2019).
>
> [2] https://github.com/zihangdai/xlnet
>
> [3] Gong, Shansan, et al. "Scaling diffusion language models via adaptation from autoregressive models." arXiv preprint arXiv:2410.17891 (2024).
>
> [4] https://huggingface.co/openai-community/gpt2
>
> [5] Sahoo, Subham, et al. "Simple and effective masked diffusion language models." Advances in Neural Information Processing Systems 37 (2024): 130136-130184.
>
> [6] Lou, Aaron, Chenlin Meng, and Stefano Ermon. "Discrete diffusion modeling by estimating the ratios of the data distribution." arXiv preprint arXiv:2310.16834 (2023).

---

### Meta-Review · Area_Chair_id59 · 2026-01-10

**Summary:**

This submission introduces Any-Subset Speculative Decoding (ASSD) for any-subset autoregressive models (AS-ARMs), a principled method for parallel token generation that still produces samples from the exact target joint distribution. The key contribution is a self-speculative accept–reject procedure with resampling, paired with formal guarantees of losslessness and a worst-case bound that the number of model evaluations never exceeds the number of generated tokens.

A major strength is that the theory cleanly fills a gap for speculative decoding beyond left-to-right autoregressive factorizations: Theorem 1 formalizes the “no NFE regret” property, and Theorem 2 establishes distributional correctness. The empirical section is consistent with the theory: it shows steady but modest speedups that grow with sequence length (roughly high-single-digit to about 11% in the reported setting), with no meaningful degradation in perplexity/entropy relative to sequential decoding.

That being said, the submission would be stronger if the practical wall-clock gains were larger and less sensitive to overhead and implementation details. As a result, the paper currently reads more as a compelling “lossless parallel decoding is possible for AS-ARMs, with strong guarantees” result than as a broadly competitive end-to-end acceleration story relative to modern, highly optimized AR decoding stacks. The added absolute comparisons and KV-caching discussion help position the contribution, but also underscore that AS-ARMs still trail optimized AR inference in wall-clock terms; the largest real-world impact likely depends on scaling and systems-level optimization beyond what is demonstrated here.

**Reviewer Concerns:**

Reviewer yrNB: The rebuttal and subsequent edits largely addressed the main "fairness of tokens" point (clarifying what token counts correspond to which training phase / initialization), and the reviewer already treated the "small speedups" concern as plausibly compute-limited. The remaining issue is not correctness but significance: the story is still that the speedups are real but modest, and the paper’s broad framing can feel slightly stronger than what the wall-clock improvements currently justify.

Reviewer BarN: The rebuttal directly addressed the scope concern by providing evidence beyond pure infilling (left-to-right generation and additional benchmark-style evaluations) and by clarifying what "accuracy" means in generative settings. What remains is mostly about depth of empirical characterization: the work now shows versatility, but still lacks a comprehensive, modern “apples-to-apples” evaluation that would convince a skeptical reader that AS-ARMs + ASSD are competitive as a practical alternative across settings (not just theoretically attractive).

Reviewer ummp: The key outstanding weakness in the original review—missing speed/perplexity comparisons to standard AR decoding (including KV caching) and masked diffusion—was addressed by adding an absolute comparison section and implementation clarifications (including how KV caching is handled). The remaining gap is largely interpretability and scope: even with added measurements, cross-model comparisons are inherently confounded by tokenizer/architecture differences, and the speedups within AS-ARMs are still not yet at the"compelling practical win" level.

**Reviewer Scores:**

Reviewer yrNB would likely stay at 8 (accept, good paper). The clarification around training-token accounting resolves the only potentially "fairness" concern, and the reviewer’s main hesitation (small speedups) was already framed as not fatal.

Reviewer BarN would likely move from 4 to around 6 (weak accept). The rebuttal meaningfully addressed "scope beyond infilling" and "accuracy analysis" with additional results and explanations, shifting the paper from "incomplete story" to "complete enough for acceptance," even if still not maximally convincing empirically.

Reviewer ummp would likely remain an 8, but with higher confidence, since the missing comparisons/clarifications were provided and the manuscript now anticipates the KV-caching and baseline-comparison questions explicitly.

Overall, after discussion and revisions, the paper clears the bar for acceptance on the strength of the guaranteed-correct, no-NFE-regret decoding result for AS-ARMs plus corroborating experiments, with the primary remaining limitation being magnitude and generality of practical speedups rather than correctness.

---

### Decision · Program_Chairs · 2026-01-26

Accept (Poster)